# Hybrid PBL and Pure PBL: Which one is more effective in developing clinical reasoning skills for general medicine clerkship?—A mixed-method study

**Kosuke Ishizuka**[ID]\*, **Kiyoshi Shikino**[ID]<sup></sup>, **Hiroki Tamura**<sup></sup>, **Daiki Yokokawa**[ID]‡, **Yasutaka Yanagita**[ID]‡, **Shun Uchida**[ID]‡, **Yosuke Yamauchi**‡, **Yasushi Hayashi**‡, **Jumpei Kojima**‡, **Yu Li**‡, **Eri Sato**‡, **Shiho Yamashita**‡, **Nao Hanazawa**‡, **Tomoko Tsukamoto**‡, **Kazutaka Noda**‡, **Takanori Uehara**‡, **Masatomi Ikusaka**‡

Department of General Medicine, Chiba University Hospital, Chiba, Japan

☯ These authors contributed equally to this work.
‡ These authors also contributed equally to this work.
\* e103007c@yokohama-cu.ac.jp

**Data Availability Statement:** All relevant data are within the paper and its Supporting Information files.

## Abstract

This study aims to compare the effectiveness of Hybrid and Pure problem-based learning (PBL) in teaching clinical reasoning skills to medical students. The study sample consisted of 99 medical students participating in a clerkship rotation at the Department of General Medicine, Chiba University Hospital. They were randomly assigned to Hybrid PBL (intervention group, n = 52) or Pure PBL group (control group, n = 47). The quantitative outcomes were measured with the students' perceived competence in PBL, satisfaction with sessions, and self-evaluation of competency in clinical reasoning. The qualitative component consisted of a content analysis on the benefits of learning clinical reasoning using Hybrid PBL. There was no significant difference between intervention and control groups in the five students' perceived competence and satisfaction with sessions. In two-way repeated measure analysis of variance, self-evaluation of competency in clinical reasoning was significantly improved in the intervention group in "recalling appropriate differential diagnosis from patient's chief complaint" ($F(1,97) = 5.295$, $p = 0.024$) and "practicing the appropriate clinical reasoning process" ($F(1,97) = 4.016$, $p = 0.038$). According to multiple comparisons, the scores of "recalling appropriate history, physical examination, and tests on clinical hypothesis generation" ($F(1,97) = 6.796$, $p = 0.011$), "verbalizing and reflecting appropriately on own mistakes," ($F(1,97) = 4.352$, $p = 0.040$) "selecting keywords from the whole aspect of the patient," ($F(1,97) = 5.607$, $p = 0.020$) and "examining the patient while visualizing his/her daily life" ($F(1,97) = 7.120$, $p = 0.009$) were significantly higher in the control group. In the content analysis, 13 advantage categories of Hybrid PBL were extracted. In the subcategories, "acquisition of knowledge" was the most frequent subcategory, followed by "leading the discussion," "smooth discussion," "getting feedback," "timely feedback," and "supporting the clinical reasoning process." Hybrid PBL can help acquire practical knowledge and deepen understanding of clinical reasoning, whereas Pure PBL can improve several important skills

**Funding:** The authors received no specific funding for this work.

**Competing interests:** The authors have declared that no competing interests exist.

such as verbalizing and reflecting on one's own errors and selecting appropriate keywords from the whole aspect of the patient.

## Introduction

Clinical reasoning is a core competency that all healthcare professionals must develop [1,2]. However, it can be challenging for beginner students because of their inadequate knowledge, poor data collection, and inappropriate approaches to information processing [1]. As such, active feedback and coaching by teachers should be tailored to individual stages of learning [3,4].

Problem-based learning (PBL) has been introduced into graduate medical education as a learner-centered educational strategy to develop clinical reasoning skills [5–8], where small groups (usually 5 to 8 people) study independently under the supervision of a tutor [5,6]. In addition, PBL using simulated patients and video materials enables free conception and direct non-verbal communication through visual and auditory information [8]. Therefore, learners can easily visualize real patients and develop the comprehensive approach required in actual clinical situations [8]. However, PBL faculty have often witnessed medical students getting stuck on a problem, which stalled the discussion [9]. In these situations, some studies have argued that a session with a faculty member can effectively solve the shortcomings of PBL [10,11].

Hybrid PBL, in which PBL is supplemented by lectures according to the student's knowledge, may be a more effective and efficient alternative to PBL alone (Pure PBL) [10,11]. While the usefulness of Hybrid PBL for clinical practice in the field of neurology has been reported, there is no study to date comparing Hybrid PBL and Pure PBL in clinical reasoning education during a general medicine clerkship [10].

The purpose of this study is thus to compare the effectiveness of Hybrid PBL and Pure PBL in teaching clinical reasoning skills to medical students.

## Methods

### Ethics statement

This research was performed per the Declaration of Helsinki and approved by the Ethics Review Committee of the Chiba University Graduate School of Medicine (Chiba, Japan). The procedures for informed consent were explained to the medical students. If consent was not obtained, the students were allowed to take the conventional Pure PBL method. Even though the researcher was also a teacher in the class, it was made clear to the medical students that this study would not affect their grade evaluations. This study has been registered in the University Hospital Medical Information Network Clinical Trials Registry (UMIN-CRT) (UMIN000045861).

### Trial design

Hybrid PBL was conducted in the intervention group, and Pure PBL was conducted in the control group. The quantitative data in this study examined students' perceived competence in PBL, satisfaction with sessions after educational intervention, and self-evaluation of competency in clinical reasoning before and after the educational intervention in the intervention and control groups. In the clinical clerkship, participants were rotated in groups assigned by

the university before this study. The participants were equally allocated to the intervention or control groups. To improve the reporting quality, we conducted a cluster randomized controlled trial based on the CONSORT 2010 statement (S1 Fig) [12]. To integrate quantitative and qualitative assessments, we designed a mixed-method sequential explanatory study [13–15]. In the intervention group, we evaluated the advantages of learning clinical reasoning via Hybrid PBL qualitatively using focus group interview (FGI) after PBL completion to examine the cognitive aspects of the medical students.

## Participants

This study was conducted at a single facility in the Department of General Medicine, Chiba University Hospital in Japan. The study included 109 fifth-year medical students at the Chiba University School of Medicine who participated in a clinical clerkship at our department from January 2020 to October 2021. This study was embedded into a clinical clerkship rotation in the department; thus, the participants were not sampled randomly. In the clinical clerkship, 5–6 medical students per group would rotate through the department every two weeks. A total of 20 groups were rotated during the study period. Each group was exposed to PBL in one case. All subjects had already received lectures and simulation training in basic and clinical medicine by the fourth year and had passed computer-based testing and objective structured clinical examinations before their clinical clerkship. Additionally, all students have experienced Pure PBL since it is integrated into Chiba University's curriculum from the first to the fourth year of medical school. Students who were unable to participate in one of the conferences for any reason were excluded from the study. We had to suspend this study because of the COVID-19 pandemic, so we only analyzed the gathered data up till then.

## Intervention

In the control group (Pure PBL group), students learn independently under the supervision of a tutor without any intervention from a faculty member [16,17]. In the intervention group (Hybrid PBL group), the tutor was allowed to help the target students understand clinical discussions beyond their own medical knowledge, to help medical students problem solve, and to answer "learner-centered" questions when the target students were stuck or overlooked the key learning issues in each scenario [18,19].

Tutors were randomly selected from 12 faculty members in our department who were PGY (postgraduate year)-4 or above (median PGY-7.5 [6–24]), with one faculty member assigned per group. Before the PBL, each teacher was provided a lecture on PBL and their role as tutors as well as some handouts. All instructions were standardized and given immediately before the PBL. Each tutor was skilled enough to explain the process of clinical reasoning to the target students.

PBL was divided into two sessions per case, with each session lasting approximately two hours (S1 Table). In the first session, history and physical examination findings were presented. In the second session, examination findings, imaging findings, and treatment plans were presented. Each group was equally assigned to one of five cases in the study groups. The five cases were Fitz-Hugh-Curtis syndrome, panic disorder, subacute combined degeneration of the spinal cord, primary amyloidosis, and sleep apnea syndrome. They present challenges in pattern recognition and can help examine analytical and diagnostic reasoning skills. In Japan, the fourth version of the national core curriculum for undergraduate medical education in 2016 introduced a new list of possible diagnoses for 37 common signs, symptoms, and pathophysiology that ought to be learned as part of the six-year undergraduate curriculum [20]. Regarding these common signs, students must acquire the competence to anticipate a set of

differential diagnoses from the earliest phase of the diagnostic process, gather confirming and refuting information according to an initial hypothesis, select and perform the relevant history taking and physical examination, and interpret the findings to confirm or deny the initial hypothesis [20,21]. The five cases were selected from those with any one of the 37 common signs that appeared on the Japanese National Medical Examination questions. A patient-simulated video and paper-based scenario assignment were prepared for each case [8]. The patient-simulated videos were produced based on the theme of case investigations. In addition to medical interview scenes in the examining rooms, they also showed the symptoms displayed in the patient's daily life. It was shown clearly to the students on a display screen. In addition, physical findings, examination findings, imaging findings, and treatment plans were distributed as paper materials.

## Outcome measures

**1) Students' perceived competence in PBL and satisfaction with sessions.**   The students' perceived competence in PBL and satisfaction with sessions after educational intervention were compared between the intervention and control groups.

After the PBL, in addition to the five students' perceived competence in PBL at the university (1. development of knowledge structure for use in clinical contexts, 2. development of an effective clinical reasoning process, 3. development of effective self-directed learning skills, 4. provision of encouragement and motivation for learning, and 5. development of team skills), item 6. satisfaction with sessions was also surveyed (S2 Table). The five students' perceived competence in PBL was determined by the faculty members of the Medical Education Laboratory of Chiba University based on the report of Barrows et al [5].

The data were collected using a self-administered 7-point Likert scale questionnaire. The scale of 7-point Likert scale was as follows: 1 = very poor, 7 = very good for students' perceived competence in PBL, and 1 = not very satisfied, 7 = very satisfied for satisfaction with sessions.

**2) Self-evaluation of competency in clinical reasoning.**   Before and after the educational intervention, we investigated the self-evaluation of competency in clinical reasoning using the following eight items: 7. recalling appropriate history, physical examination, and tests on clinical hypothesis generation, 8. recalling appropriate differential diagnosis from the patient's chief complaint, 9. verbalizing points that fit/don't fit the recalled differential diagnosis appropriately, 10. verbalizing and reflecting appropriately on own mistakes, 11. selecting keywords from the whole aspect of the patient, 12. examining the patient while visualizing his/her daily life, 13. considering biological, psychological, and social perspectives, and 14. practicing the appropriate clinical reasoning process (S2 Table). The self-evaluation of competency in clinical reasoning investigated in this study was based on the report of Cooper et al. and was determined through discussions among the faculty members of our department [3].

The data were collected using a self-administered 7-point Likert scale questionnaire. The scale of the 7-point Likert scale was as follows: 1 = not at all confident, 7 = very confident for self-evaluation of competency in clinical reasoning.

## Sample size

Because this study also served as an educational program for a clinical clerkship in our department, medical students who were rotated at the beginning of the study and all medical students who would rotate thereafter in the same year were included in the study. For the quantitative data, the sample size required for a two-tailed t-test of the difference between the means of the two groups was calculated to be 128 students in total (64 students in each group), assuming a significance level of 0.05, a power of 0.8, and an effect size of 0.5. When the Mann-Whitney U

test was conducted at the significance level of 0.05, power of 0.8, and effect size of 0.5, the required sample size was calculated to be 67 persons in each group, for a total of 134 persons. However, we had to suspend this study because of the COVID-19 pandemic and only analyzed the gathered data till then. Therefore, a total of 109 students across 20 groups were considered.

## Randomization

The 20 eligible groups were allocated to the intervention or control groups in an equal number of groups. The groups were allocated through cluster randomization using Microsoft Excel (Microsoft Co., Redmond, WA, USA). The allocation was not blinded to the medical students or the faculty.

## Statistical method

All statistical analyses were performed using SPSS Statistics for Windows 26.0 (IBM Co., Armonk, NY, USA) with a significance level of less than 5%. Sex and age were analyzed using the chi-square test and Mann-Whitney U test respectively. The results of the questionnaires on students' perceived competence in PBL and satisfaction with sessions after educational intervention were compared between the intervention and control groups using an unpaired t-test (Fig 1). Regarding self-evaluation of competency in clinical reasoning, we conducted an analysis of covariance (ANCOVA) between intervention and control groups before educational intervention as a covariate and the method of PBL as a fixed factor. In addition, to examine whether the self-evaluation of competency in clinical reasoning changed between the intervention and control groups, we conducted a two-way repeated measure analysis of variance (ANOVA), where the within-subjects factor was the time of implementation, and the between-subjects factor was the method of PBL.

## Focus group interview

Qualitative evaluation was conducted following the quantitative evaluation to assess the acquisition of higher-order intellectual skills, and the results were integrated as a mixed-method sequential explanatory study [13–15]. Because all of the target students had experienced Pure PBL by their fourth year but not Hybrid PBL, the medical students in the intervention group were selected as the sampling group for our qualitative survey. We evaluated the advantages of learning clinical reasoning via Hybrid PBL qualitatively using focus group interview (FGI) after PBL completion to examine the cognitive aspects of the medical students, which is thought to influence the learning effectiveness of clinical reasoning by PBL. After obtaining informed consent from medical students in the intervention group, we conducted an FGI face-to-face immediately after completion of the PBL, which lasted about 10 minutes to minimize participants' fatigue. In the FGI, the advantages of Hybrid PBL in improving clinical reasoning skills were investigated. The aims of the study were reviewed at the beginning of each focus group. The interview content was discussed among the interviewers (HT, KS), and an interview guide was developed (S3 Table). Medical students were asked the following open-ended questions: "What are the advantages of Hybrid PBL? Why do you think so?" The responses of the focus groups were recorded and transcribed verbatim. One faculty member administered the FGI to a group of 5–6 medical students immediately after filling out the post-PBL questionnaires. The FGI sampling was 52 students (10 groups) from all intervention groups. The FGI interviewers were trained facilitators from the faculty (KI, KS, YH, YY, JK, SU, YY, DY, ES, NH, SY, TT, KN) who were in charge of PBL. They had experience in higher education in their respective countries and had previously conducted educational research. The FGI interviewers did not share their personal attitudes and behaviors. A team debrief was

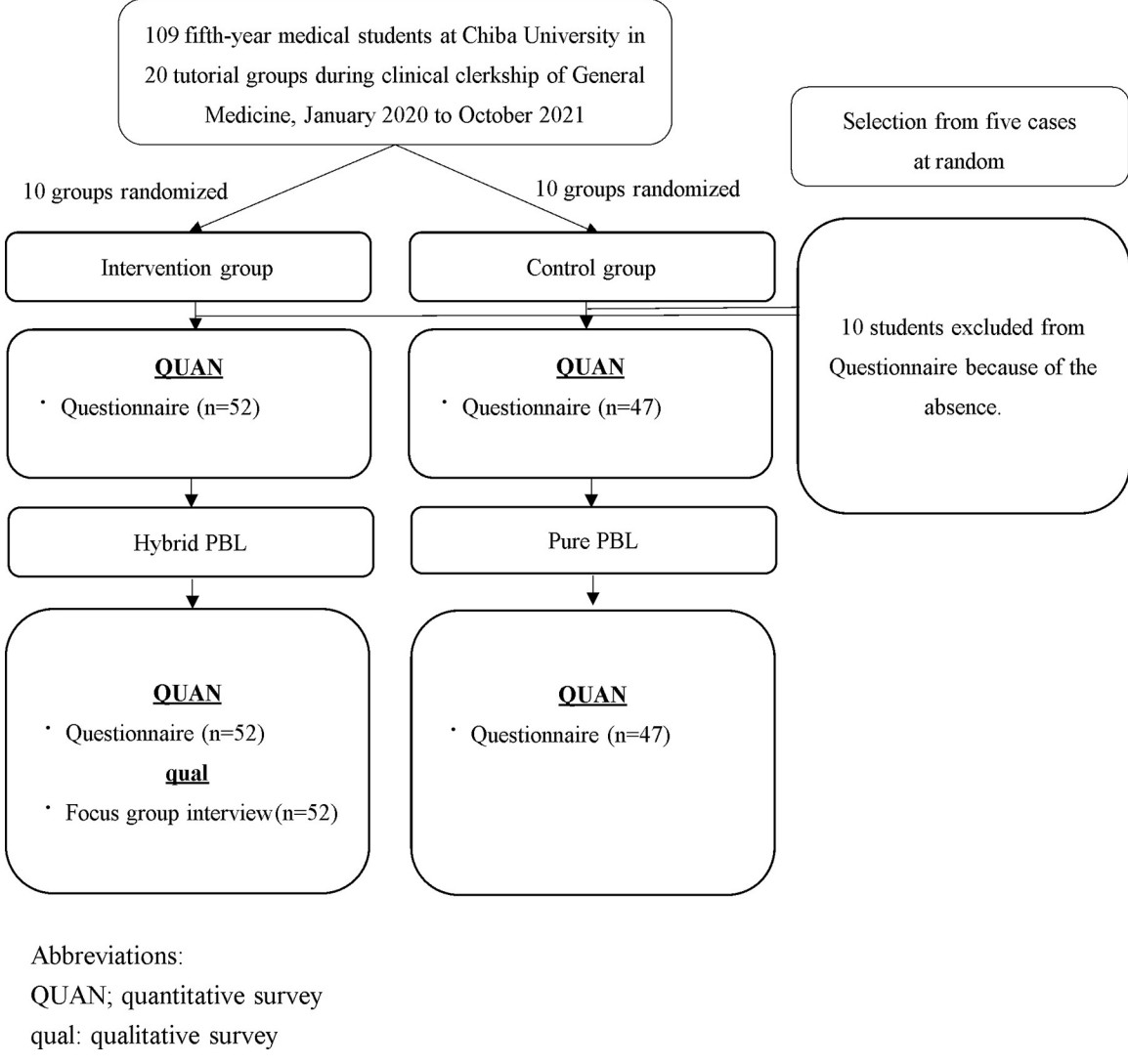

Abbreviations:

QUAN; quantitative survey

qual: qualitative survey

**Fig 1. Flow diagram of the design.**

held after each focus group session, and field notes were documented. There were no repeat interviews. Participants were not asked for feedback nor to review transcripts.

For the qualitative research, content analysis was used to analyze the themes (Table 1) [22,23]. A preliminary analytic template, aligned with the focus group guide, was developed as a starting point for analysis. Two researchers (KI, HT) independently read and did the initial coding of all focus group transcripts. Thereafter, to ensure the quality of the research, researcher triangulation was conducted by two researchers (KI, HT), who discussed, identified, and agreed on the coding of the descriptors. Following the coding, similar codes were grouped into categories and subcategories, which were regularly discussed and reviewed by KS (who had experience in qualitative research) to ensure the credibility of the findings. Cohen's kappa coefficient was used to evaluate the intra-rater validity between the two researchers (0.8–1.0 = almost perfect, 0.6–0.8 = substantial, 0.4–0.6 = moderate, 0.2–0.4 = good) [24]. The consolidated criteria for reporting qualitative research (COREQ) checklist was used to report our findings [25].

**Table 1. The step of qualitative content analysis.**

| Hybrid PBL approach | |
|---|---|
| **Step** | **Description** |
| **Step 1: Overview** | A preliminary analytic template, aligned with the focus group guide, was developed as a starting point for analysis.<br>All 52 medical students in the intervention group were included in the qualitative analysis.<br>General themes were identified.<br>A coding system was developed. |
| **Step 2: Open and axial coding** | Open coding: A smaller sample of the documents was read. One randomly selected record was analyzed. Noticeable patterns were recorded. Emerging and predetermined codes were used during analysis.<br>Axial coding: Entire sample of documents was reviewed. Specific passages belonging under theme categories identified in initial open coding were tagged. |
| **Step 3: Selective- coding** | The researcher combed through the documents in search of miscoded passages and discrepant evidence. |
| **Step 4: Independent analysis** | Two researchers (KI, HT) independently read and did the coding of all focus group transcripts. |
| **Step 5: Description** | Descriptors as well as themes for analysis were compiled.<br>The coding system was then used to generate themes for the study.<br>The themes were linked to the major findings of the study.<br>The findings were linked to multiple perspectives and specific evidence. |
| **Step 6: Discussion of themes** | To ensure the quality of the research, researcher triangulation was conducted by two researchers (KI, HT), who discussed, identified, and agreed on the coding of the descriptors. |
| **Step 7: Interpretation and verification** | Interpretation of meaning derived from the study.<br>Following the coding, similar codes were grouped into categories and subcategories, which were regularly discussed and reviewed by KS (who had experience in qualitative research) to ensure the credibility of the findings. |
| **Step 8: Comparison and theory** | Comparison with relevant literature and theory was done.<br>Implications for future research and reform were outlined. |

To examine the effects of Hybrid PBL on the cognitive processes of medical students, the analytic categories were set according to the six cognitive process dimensions of the revised Bloom's taxonomy [26]. After open coding, similar codes were classified into subcategories and categories. We analyzed the concepts in each of the six cognitive process dimensions in the revised Bloom's taxonomy and calculated the number of analysis units for each concept [26]. We also grouped similar codes as themes and checked which dimension of the cognitive process dimension they correspond to.

## Mixed methods research

To integrate quantitative and qualitative assessments, a mixed methods study was conducted with exploratory sequential design [13–15]. This type of research study design capitalizes on the respective strengths of both quantitative and qualitative design while minimizing each other's shortcomings. Furthermore, it allows the researchers to understand the experimental results better while incorporating the medical students' perspectives. This is based on the advice of The National Institute of Health, which advocates a mixed-method approach to research "to improve the quality and scientific power of data" and to better address the complexity of issues in health science education today [22,27].

## Results

### Participant characteristics

Consent was obtained from all 109 participating students. Ninety-nine students (90.8%) were included in the quantitative evaluation, while ten students who missed at least one PBL session

**Table 2. Participant characteristics.**

|  | Total (n = 99) | Hybrid PBL (n = 52) | Pure PBL (n = 47) | P Value |
|---|---|---|---|---|
| Male | 75 (75.8%) | 39 (75.0%) | 36 (76.6%) | p = 0.853 |
| Age, average (SD) | 22.8 (1.6) | 22.6 (1.3) | 23.0 (1.8) | p = 0.085 |

were excluded. 52 participants (10 groups) were assigned to the intervention group and 47 participants (10 groups) to the control group. Comparison between the two groups showed no significant difference in age and sex (Table 2).

**1) Students' perceived competence in PBL and satisfaction with sessions.** Students' perceived competence in PBL after educational intervention and satisfaction with sessions were not significantly different between two groups (Table 3).

**2) Self-evaluation of competency in clinical reasoning.** Table 4 showed the baseline of self-evaluation of competency in clinical reasoning before educational intervention between the intervention and the control groups. Self-evaluation of competency in clinical reasoning before the PBL: 7. recalling appropriate history, physical examination, and tests on clinical hypothesis generation ($3.04 \pm 1.27$ vs. $2.42 \pm 0.92$, p = 0.006), 8. recalling appropriate differential diagnosis from patient's chief complaint ($3.06 \pm 1.28$ vs. $2.54 \pm 0.96$, p = 0.022), 9. verbalizing points that fit/don't fit the recalled differential diagnosis appropriately ($3.15 \pm 1.32$ vs. $2.65 \pm 1.03$, p = 0.039), 14. practicing the appropriate clinical reasoning process ($2.91 \pm 1.25$ vs. $2.31 \pm 0.96$, p = 0.008) were significantly higher in the control group. There was no significant difference between the intervention and the control groups in the following items: 10. verbalizing and reflecting appropriately on own mistakes, 11. selecting keywords from the whole aspect of the patient, 12. examining the patient while visualizing his/her daily life, and 13. considering biological, psychological, and social perspectives.

In ANCOVA of the self-evaluation of competency in clinical reasoning after the educational intervention, there was no significant difference between the intervention and the control groups in all items (Table 5).

**Table 3. Fifth-year medical students' perceived competence in PBL and satisfaction with sessions after educational intervention, about hybrid- and pure -PBL, Chiba University Hospital (N = 99).**

| Questionnaire (7-point Likert scale) | PBL | Mean ± SD | p value |
|---|---|---|---|
| 1. Development of knowledge structure for use in clinical contexts | Hybrid | 5.00 ± 1.16 | 0.918 |
|  | Pure | 5.02 ± 0.85 |  |
| 2. Development of an effective clinical reasoning process | Hybrid | 5.44 ± 0.96 | 0.323 |
|  | Pure | 5.62 ± 0.77 |  |
| 3. Development of effective self-directed learning skills | Hybrid | 5.06 ± 1.00 | 0.387 |
|  | Pure | 4.89 ± 0.87 |  |
| 4. Provision of encouragement and motivation for learning | Hybrid | 5.62 ± 0.91 | 0.746 |
|  | Pure | 5.55 ± 1.00 |  |
| 5. Development of team skills | Hybrid | 5.27 ± 1.12 | 0.654 |
|  | Pure | 5.17 ± 1.07 |  |
| 6. Satisfaction with sessions | Hybrid | 5.35 ± 0.86 | 0.858 |
|  | Pure | 5.38 ± 1.17 |  |

(7-point Likert scale rating of students' perceived competence in PBL: 1 = very poor, 7 = very good, and scale rating of satisfaction with sessions: 1 = very dissatisfied, 7 = very satisfied).

**Table 4. Baseline of fifth-year medical students' self-evaluation of competency in clinical reasoning before the educational intervention, based on the 7-point Likert scale about hybrid- and pure-PBL, Chiba University Hospital (N = 99).**

| Questionnaire (7-point Likert scale) | Confidence (Mean ± SD) | | P Value |
|---|---|---|---|
| | **Hybrid PBL** | **Pure PBL** | |
| 7. Recalling appropriate history, physical examination, and tests on clinical hypothesis generation | 2.42 ± 0.92 | 3.04 ± 1.27 | p = 0.006* |
| 8. Recalling appropriate differential diagnosis from patient's chief complaint | 2.54 ± 0.96 | 3.06 ± 1.28 | p = 0.022* |
| 9. Verbalizing points that fit/don't fit the recalled differential diagnosis appropriately | 2.65 ± 1.03 | 3.15 ± 1.32 | p = 0.039* |
| 10. Verbalizing and reflecting appropriately on own mistakes | 3.04 ± 1.17 | 3.38 ± 1.21 | p = 0.153 |
| 11. Selecting keywords from the whole aspect of the patient | 2.98 ± 0.87 | 3.36 ± 1.21 | p = 0.073 |
| 12. Examining the patient while visualizing his/her daily life | 2.92 ± 1.01 | 3.32 ± 1.18 | p = 0.075 |
| 13. Considering biological, psychological, and social perspectives | 2.87 ± 0.93 | 3.06 ± 1.05 | p = 0.321 |
| 14. Practicing the appropriate clinical reasoning process | 2.31 ± 0.96 | 2.91 ± 1.25 | p = 0.008* |

(7-point Likert scale rating of self-evaluation of competency: 1 = not at all confident, 7 = very confident).

In two-way ANOVA of self-evaluation of competency in clinical reasoning, a significant interaction effect was shown in item 8. recalling appropriate differential diagnosis from the patient's chief complaint ($F(1,97) = 5.295$, $p = 0.024$) and item 14. practicing the appropriate clinical reasoning process ($F(1,97) = 4.016$, $p = 0.038$) (Table 6). There was no significant interaction effect between the intervention and the control groups in items 7. recalling appropriate history, physical examination, and tests on clinical hypothesis generation, 9. verbalizing points that fit/don't fit the recalled differential diagnosis appropriately, 10. verbalizing and reflecting appropriately on own mistakes, 11. selecting keywords from the whole aspect of the patient, 12. examining the patient while visualizing his/her daily life, and 13. considering biological, psychological, and social perspectives (Table 6). According to multiple comparisons, the control group showed significantly better results in item 7. recalling appropriate history, physical examination, and tests on clinical hypothesis generation ($F(1,97) = 6.796$, $p = 0.011$), item 10. verbalizing and reflecting appropriately on own mistakes ($F(1,97) = 4.352$, $p = 0.040$), item 11.

**Table 5. ANCOVA in fifth-year medical students' self-evaluation of competency in clinical reasoning after the educational intervention based on the 7-point Likert scale about hybrid- and pure-PBL, Chiba University Hospital (N = 99).**

| Questionnaire (7-point Likert scale) | Post-PBL confidence adjusted for baseline (Mean ± SD) | | Difference between average value | 95% confidence interval | P value | Effect size |
|---|---|---|---|---|---|---|
| | **Hybrid PBL** | **Pure PBL** | | | | |
| 7. Recalling appropriate history, physical examination, and tests on clinical hypothesis generation | 4.45 ± 0.15 | 4.59 ± 0.16 | −0.14 | −0.57 to 0.30 | p = 0.541 | 0.420 |
| 8. Recalling appropriate differential diagnosis from patient's chief complaint | 4.51 ± 0.15 | 4.27 ± 0.16 | 0.24 | -0.20 to 0.69 | p = 0.282 | 1.375 |
| 9. Verbalizing points that fit/don't fit the recalled differential diagnosis appropriately | 4.86 ± 0.14 | 4.86 ± 0.15 | 0.01 | -0.41 to 0.42 | p = 0.977 | 0.001 |
| 10. Verbalizing and reflecting appropriately on own mistakes | 4.60 ± 0.13 | 4.91 ± 0.14 | −0.30 | -0.68 to 0.08 | p = 0.120 | 2.196 |
| 11. Selecting keywords from the whole aspect of the patient | 4.68 ± 0.14 | 4.99 ± 0.15 | -0.31 | -0.73 to 0.10 | p = 0.139 | 2.330 |
| 12. Examining the patient while visualizing his/her daily life | 4.61 ± 0.13 | 4.99 ± 0.14 | -0.39 | -0.78 to 0.00 | p = 0.052 | 3.549 |
| 13. Considering biological, psychological, and social perspectives | 4.72 ± 0.93 | 4.84 ± 0.14 | -0.13 | -0.51 to 0.26 | p = 0.517 | 0.383 |
| 14. Practicing the appropriate clinical reasoning process | 4.41 ± 0.15 | 4.31 ± 0.16 | 0.10 | -0.33 to 0.53 | p = 0.644 | 0.235 |

(7-point Likert scale rating of self-evaluation of competency: 1 = not at all confident, 7 = very confident).

**Table 6. A two-way ANOVA in fifth-year medical students' self-evaluation of competency in clinical reasoning before and after the educational intervention based on the 7-point Likert scale, about hybrid- and pure-PBL, Chiba University Hospital (N = 99).**

| Questionnaire (7-point Likert scale) | PBL | Confidence (Mean ± SD) | | F(p value) | | |
|---|---|---|---|---|---|---|
| | | Pre-PBL | Post-PBL | The method of PBL | The time of implementation of PBL | Effect modification |
| 7. Recalling appropriate history, physical examination, and tests on clinical hypothesis generation | Hybrid | 2.42 ± 0.92 | 4.38 ± 1.07 | 6.796 (p = 0.011*) | 173.224 (p<0.001*) | 1.606 (p = 0.208) |
| | Pure | 3.04 ± 1.27 | 4.66 ± 1.09 | | | |
| 8. Recalling appropriate differential diagnosis from patient's chief complaint | Hybrid | 2.54 ± 0.96 | 4.44 ± 1.06 | 1.393 (p = 0.241) | 136.138 (p<0.001*) | 5.295 (p = 0.024*) |
| | Pure | 3.06 ± 1.28 | 4.34 ± 1.19 | | | |
| 9. Verbalizing points that fit/don't fit the recalled differential diagnosis appropriately | Hybrid | 2.65 ± 1.03 | 4.83 ± 1.08 | 2.769 (p = 0.099) | 189.088 (p<0.001*) | 2.261 (p = 0.136) |
| | Pure | 3.15 ± 1.32 | 4.89 ± 0.94 | | | |
| 10. Verbalizing and reflecting appropriately on own mistakes | Hybrid | 3.04 ± 1.17 | 4.58 ± 0.98 | 4.352 (p = 0.040*) | 126.778 (p<0.001*) | 0.003 (p = 0.957) |
| | Pure | 3.38 ± 1.21 | 4.94 ± 0.94 | | | |
| 11. Selecting keywords from the whole aspect of the patient | Hybrid | 2.98 ± 0.87 | 4.65 ± 1.10 | 5.607 (p = 0.020*) | 149.420 (p<0.001*) | 0.002 (p = 0.961) |
| | Pure | 3.36 ± 1.21 | 5.02 ± 0.94 | | | |
| 12. Examining the patient while visualizing his/her daily life | Hybrid | 2.92 ± 1.01 | 4.60 ± 1.00 | 7.120 (p = 0.009*) | 139.097 (p<0.001*) | 0.001 (p = 0.978) |
| | Pure | 3.32 ± 1.18 | 5.00 ± 0.91 | | | |
| 13. Considering biological, psychological, and social perspectives | Hybrid | 2.87 ± 0.93 | 4.69 ± 0.96 | 1.420 (p = 0.236) | 231.484 (p<0.001*) | 0.006 (p = 0.939) |
| | Pure | 3.06 ± 1.05 | 4.87 ± 0.99 | | | |
| 14. Practicing the appropriate clinical reasoning process | Hybrid | 2.31 ± 0.96 | 4.35 ± 1.06 | 5.120 (p = 0.064) | 167.653 (p<0.001*) | 4.016 (p = 0.038*) |
| | Pure | 2.91 ± 1.25 | 4.38 ± 1.07 | | | |

(7-point Likert scale rating of self-evaluation of competency: 1 = not at all confident, 7 = very confident).

selecting keywords from the whole aspect of the patient (F(1,97) = 5.607, p = 0.020), and item 12. examining the patient while visualizing his/her daily life (F(1,97) = 7.120, p = 0.009) (Table 6). In addition, the intervention and control groups showed significant improvement in self-evaluation of competency in clinical reasoning after the implementation of PBL (Table 6).

## Content analysis

All 52 medical students (94.5%) in the intervention group consented to participate in the FGI immediately after the completion of the PBL and were included in the questionnaire analysis. After analyzing the records of the 10 focus groups, we confirmed that we had reached thematic saturation. Table 7 shows the categories, subcategories, number of codes, and representative quotations in the present data. Regarding the advantages of the intervention group, we identified 13 categories and 27 subcategories that correspond to the four stages of the revised Bloom's taxonomy (remember, understand, apply, and analyze) [26]. The most frequent cognitive domain in the revised Bloom's taxonomy was "understand." In the subcategories of content analysis, "acquisition of knowledge" was the most frequent subcategory, followed by "leading the discussion," "smooth discussion," "getting feedback," "timely feedback," and "supporting the clinical reasoning process." Interrater reliability was substantial (Cohen's kappa = 0.81).

**Table 7. Absolute frequencies of codes for each theme (intervention group).**

| | Hybrid PBL approach | | | |
|---|---|---|---|---|
| Cognitive process levels from the revised Bloom's Taxonomy | Category | Subcategory | Code | Quotes |
| **Analyze (7)** | Complementing one's own experience (5) | Supporting the clinical reasoning process (5) | Not having enough experience in clinical reasoning | "I think that the medical students do not have enough experience in clinical reasoning." |
| | Curiosity (2) | Increasing motivation (2) | Feeling more at ease speaking up | "I believe that the medical students feel more at ease speaking up in hybrid PBL because the teacher is allowed to speak up." |
| **Apply (37)** | Smooth discussion and understanding (31) | Leading the discussion (18) | Leading the discussion | "I feel that the advantage of hybrid PBL is that the teacher can lead the discussion without delay." |
| | | Smooth discussion (12) | Smooth discussion | "One advantage of hybrid PBL is the smooth discussion with the teacher during the clinical reasoning process, in my opinion." |
| | | Active discussion (1) | Easily starting a discussion | "I feel that the advantage of hybrid PBL is that when medical students are stuck on a problem, they can easily start a discussion by asking the teacher for guidance." |
| | Application to problem-solving (3) | Solving the difficult problem (2) | Being guided even with difficult problems | "I like hybrid PBL because it can be guided in the right direction even with difficult problems." |
| | | Applying knowledge (1) | Applying the concepts of clinical reasoning | "I am confident I can apply the concepts of clinical reasoning when I am faced with a real case thanks to hybrid PBL." |
| | Psychological safety (2) | Psychological safety (2) | Psychological safety | "I like the psychological safety from hybrid PBL because medical students get advice from the teacher." |
| | Self-regulated learning (1) | Establishing a self-learning style (1) | Establishing a more self-oriented learning style | "I believe that hybrid PBL allows medical students to establish a more self-oriented learning style." |
| **Understand (43)** | Effective feedback (21) | Getting feedback (12) | Getting feedback | "The major advantage of Hybrid PBL is that medical students can get feedback on the clinical reasoning process from the teacher." |
| | | Timely feedback (7) | Getting the teacher's knowledge in real-time | "I like that hybrid PBL allows medical students to get the teacher's knowledge in real-time." |
| | | Appropriate feedback for errors (1) | Correcting the misdiagnosis | "I enjoy the fact that hybrid PBL allows the teacher can correct the misdiagnosis of medical students." |
| | | Clarifying learning points (1) | Clarifying learning points | "I think it is great that hybrid PBL can clarify learning points." |
| | Understanding clinical reasoning (7) | Understanding the clinical reasoning process (3) | Understanding the correct clinical reasoning process | "I feel that hybrid PBL allows medical students to understand the correct clinical reasoning process." |
| | | Understanding the process of recalling the disease (2) | Understanding the process of recalling the disease | "After going through hybrid PBL, I think medical students can better understand the process of recalling the disease." |
| | | Considering the points that fit and those that don't (1) | Empowering appropriate consideration of the points that fit/ don't fit the recalled disease | "I think that hybrid PBL empowers appropriate consideration of the points that fit/don't fit the recalled disease." |
| | | Understanding the framework of clinical reasoning (1) | Understand the correct clinical reasoning process | "An advantage of hybrid PBL is how it allows medical students to understand the correct clinical reasoning process." |
| | Deep understanding (6) | Being a learning experience (3) | More effective learning experience | "I felt overall that hybrid PBL is a more effective learning experience." |
| | | Avoiding oversights (2) | Avoid overlooking important clinical information | "One good thing about hybrid PBL is that it avoids overlooking important clinical information." |
| | | Understanding the process from a broad perspective (1) | Understanding the process from a broad perspective | "I liked that hybrid PBL allows medical students to understand the process from a broad perspective." |
| | Efficient teaching (5) | Quickly learning (3) | Early acquisition of knowledge | "I think one advantage of hybrid PBL is the early acquisition of knowledge." |
| | | Time efficiency (1) | Saving a lot of time | "I think hybrid PBL saves a lot of time." |
| | | Time savings (1) | Not wasting any time | "Hybrid PBL does not waste any time." |
| | The right direction (4) | Correcting the course of the discussion (4) | Correcting the course of the discussion | "I liked that hybrid PBL allows medical students to correct the course of the discussion." |
| **Remember (22)** | Acquiring knowledge (21) | Acquiring knowledge (20) | Acquisition of accurate knowledge | "I believe that one advantage of hybrid PBL is the acquisition of accurate knowledge." |
| | | Reacquiring knowledge (1) | Reacquisition of accurate knowledge | "I felt like hybrid PBL helped me reacquire accurate knowledge." |
| | Memory retention (1) | Memory retention (1) | Retaining the knowledge | "I think that hybrid PBL improves the ability to retain the knowledge." |

*() number of codes.

## Discussion

This study suggests that Hybrid PBL and Pure PBL have their own advantages in helping medical students to acquire clinical reasoning skills. There was no significant difference between the intervention and the control groups in the five students' perceived competence in PBL and satisfaction with sessions. In ANCOVA of the self-evaluation of competency in clinical reasoning after the educational intervention, there was no significant difference between the intervention and the control groups in all items. Meanwhile, the two-way ANOVA of the self-evaluation of competency in clinical reasoning before and after the educational intervention showed that the intervention group significantly improved in items 8. recalling appropriate differential diagnosis from the patient's chief complaint and 14. practicing the appropriate clinical reasoning process. However, according to multiple comparisons, the control group still scored significantly higher than the intervention group in items 7. recalling appropriate history, physical examination, and tests on clinical hypothesis generation, 10. verbalizing and reflecting appropriately on own mistakes, 11. selecting keywords from the whole aspect of the patient, and 12. examining the patient while visualizing his/her daily life. The difference in results between the ANCOVA and the two-way ANOVA in self-evaluation of competency in clinical reasoning was considered to be smaller number of participants and baseline differences.

There was no significant difference between the intervention and the control groups in the five students' perceived competence in PBL and satisfaction with sessions. It is suggested that the intervention group achieved the original goals of PBL reported by Barrows et al. to the same extent as the control group [5].

The two-way ANOVA of the self-evaluation of competency in clinical reasoning before and after the educational intervention showed that the intervention group significantly improved in items 8. recalling appropriate differential diagnosis from the patient's chief complaint and 14. practicing the appropriate clinical reasoning process. Furthermore, in the subcategories of content analysis, "acquisition of knowledge" was the most frequent subcategory, followed by "leading the discussion," "smooth discussion," "getting feedback," "timely feedback," and "supporting the clinical reasoning process." This indicates that acquiring practical knowledge and deepening understanding through timely and effective feedback may be an advantage of Hybrid PBL. The disadvantages of Pure PBL are that it is ineffective for medical students with low self-learning ability, it increases the variability of learning outcomes among medical students, it is difficult to have effective discussions when medical students do not understand the concept of clinical reasoning, it does not allow students to learn knowledge systematically, it does not provide deep and practical knowledge, and students tend to miss important topics [10]. From the qualitative research of this study, those categories and codes were extracted as advantages of Hybrid PBL. The integration of quantitative and qualitative data signifies an advantage of Hybrid PBL in that it enhances the acquisition of practical knowledge and understanding in the process of clinical reasoning through timely and effective feedback from the tutors.

Based on the two-way ANOVA of self-evaluation of competency in clinical reasoning, the control group scored higher than the intervention group in items 7. recalling appropriate history, physical examination, and tests on clinical hypothesis generation, 10. verbalizing and reflecting appropriately on own mistakes, 11. selecting keywords from the whole aspect of the patient, and 12. examining the patient while visualizing his/her daily life. PBL not only enhances problem-solving skills in clinical situations but also encourages medical students to learn independently [5]. The advantage of Pure PBL is that medical students can independently have the time and process to think carefully, integrate and interpret the information

based on the collected information, and verbalize and reflect on the errors in the process. Therefore, Pure PBL can help improve several important skills in clinical reasoning, such as verbalizing and reflecting on the process of one's own errors and selecting the appropriate keywords from the whole aspect.

## Limitations

The limitations of this study are as follows:

First, the differences in results between the ANCOVA and the two-way ANOVA in self-evaluation of competency in clinical reasoning were considered to be smaller number of participants and baseline differences. Sampling bias may have occurred between both groups. In addition, an alpha error may have occurred in the statistical analysis of the two-way ANOVA because it did not follow a normal distribution. Owing to the coronavirus disease 2019 pandemic, this study was suspended and resumed. The results should be accepted cautiously as this is a single study with relatively few participants.

Second, this study was conducted using scenario tasks with patient-simulated video and paper-based scenarios, not actual patients. It is necessary to verify whether the advantages of Hybrid PBL are similar to those of Pure PBL for real patients.

Third, the effectiveness of clinical reasoning education by Hybrid PBL may vary depending on the teaching skills of the teachers. To control for this, the instruction and training of teachers were designed to minimize the influence of their instructional skills.

Fourth, this study was conducted on fifth-year medical students at a single institution. Therefore, the results of this study may not be generalizable or transferable beyond the specific population from which the sample was drawn. Further validation is needed to determine whether the results can be applied to residents and general physicians.

Fifth, since we only held the focus groups among the intervention group, it could have had a separate effect on students' competence in clinical reasoning via reflection or simply increased interaction with the tutors. Also, this might have introduced bias as a co-intervention in the intervention group.

Sixth, the allocation to the two groups could not be blinded to the participants and the faculty members, and subjective bias may have affected the results. Furthermore, the assignment of these groups by the university before the start of this study might also have been biased, which may have affected the study results.

## Conclusion

This study suggests that Hybrid PBL and Pure PBL have their own advantages in teaching medical students to acquire clinical reasoning skills. Hybrid PBL can better help acquire practical knowledge and deepen understanding in the process of clinical reasoning through timely and effective feedback from the tutors, whereas Pure PBL is more useful for improving several important skills in the process of clinical reasoning, such as verbalizing and reflecting on the process of one's own errors and selecting appropriate keywords from the whole aspect.

## Supporting information

**S1 Fig. CONSORT 2010 flow diagram.**
(TIF)

**S1 Table. The outline of the PBL flow.**
(PDF)

**S2 Table. The questionnaire of the Fifth-year medical students' perceived competence in PBL, satisfaction with sessions, and self-evaluation of competency in clinical reasoning about hybrid- and pure-PBL, Chiba University Hospital (N = 99).**
(PDF)

**S3 Table. Interview guidelines.**
(PDF)

## Acknowledgments

The authors thank the physicians who participated in the present study.

## Author Contributions

**Conceptualization:** Kosuke Ishizuka.

**Data curation:** Kosuke Ishizuka, Kiyoshi Shikino, Hiroki Tamura.

**Formal analysis:** Kosuke Ishizuka, Kiyoshi Shikino, Hiroki Tamura.

**Investigation:** Kosuke Ishizuka, Kiyoshi Shikino, Hiroki Tamura, Daiki Yokokawa, Yasutaka Yanagita, Shun Uchida, Yosuke Yamauchi, Yasushi Hayashi, Jumpei Kojima, Yu Li, Eri Sato, Shiho Yamashita, Nao Hanazawa, Tomoko Tsukamoto, Kazutaka Noda, Takanori Uehara.

**Methodology:** Kosuke Ishizuka, Kiyoshi Shikino.

**Project administration:** Kosuke Ishizuka, Kiyoshi Shikino.

**Resources:** Kosuke Ishizuka.

**Software:** Kosuke Ishizuka.

**Supervision:** Kiyoshi Shikino, Masatomi Ikusaka.

**Validation:** Kosuke Ishizuka.

**Visualization:** Kosuke Ishizuka.

**Writing – original draft:** Kosuke Ishizuka.

**Writing – review & editing:** Kiyoshi Shikino, Hiroki Tamura, Daiki Yokokawa, Yasutaka Yanagita, Shun Uchida, Yosuke Yamauchi, Yasushi Hayashi, Jumpei Kojima, Yu Li, Eri Sato, Shiho Yamashita, Nao Hanazawa, Tomoko Tsukamoto, Kazutaka Noda, Takanori Uehara, Masatomi Ikusaka.

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
