## [Decision Letter · Decision Letter 0]

9 Aug 2022

PONE-D-22-20122Hybrid PBL and Pure PBL: Which one is more effective in developing clinical reasoning skills for general medicine clerkship? - A mixed-methods randomized control trialPLOS ONE

Dear Dr. Ishizuka,

Thank you for submitting your manuscript to PLOS ONE. After careful consideration, we feel that it has merit but does not fully meet PLOS ONE’s publication criteria as it currently stands. Therefore, we invite you to submit a revised version of the manuscript that addresses the points raised during the review process.

We look forward to receiving your revised manuscript.

Kind regards,

Somayeh Delavari, Ph.D.,

Academic Editor

PLOS ONE

Journal Requirements:

2. Please remove your figures from within your manuscript file, leaving only the individual TIFF/EPS image files, uploaded separately.  These will be automatically included in the reviewers’ PDF.

Additional Editor Comments:

Reviewer 1:

General Comments:

The authors aimed to test whether a hybrid PBL approach that adds a lecture session to pure PBL, enhanced the medical students’ clinical reasoning competencies compared to pure PBL. This is an important and novel question.

Their approach to the problem is particularly interesting since they have adopted a combination of a randomized controlled trial design with a qualitative focus group to assess the in-depth experience of the learners. Moreover, the authors have assessed the metacognitive aspects of clinical reasoning among their outcomes. The video supplement to the PBL sessions that included the patient’s lifestyle also grabbed my attention.

However, there are some major concerns regarding this piece of research and the way it’s presented that need to be addressed before considering the manuscript for publication in PLOS One.

• The manuscript is lengthy and this has buried its interesting and important aspects. It can be summarized to at least half its current length without sacrificing important points.

• There are major concerns about the clarity of some parts of the text.

• The authors should clarify some methodological aspects detailed below but namely

o their approach to randomization has not been described,

o the high number of outcomes that may increase the likelihood of a type I error, and how the authors have adjusted for the effects of baseline difference,

o And that the actual sample size was smaller than the calculated sample size and this may pose questions regarding the reliability of the negative findings.

Introduction:

The introduction is hard to follow and contains unnecessary explanations. I suggest summarizing it by only including a few sentences on the brief description of clinical reasoning and its significance, the components of competency in clinical reasoning, the role of PBL, and its deficits in clinical reasoning development in about two paragraphs. Followed by a third paragraph on the definition of hybrid PBL and how it can solve current problems concerning directly relevant research.

• The term “zone of proximal development” on line 99 is unclear and confusing to me. I guess the authors mean that the learners’ baseline knowledge and past experiences should be taken into account when designing a PBL session.

• There are several paragraphs in the introduction that seem redundant and only bury the main message in a lengthy discussion of unnecessary information.

Methods:

The RCT methodology is an appropriate method for the authors’ research question. However, some issues need to be addressed. The main concerns are about the randomization methods and the composition of the manuscript.

• The presentation of the methods section can be improved so that it is easier to follow. I suggest using subsections for the “participants and trial design” section: The design: where you describe the study briefly; The Participants and setting: where you describe your participants and their background including the details of the medical education system at your institution with special attention to PBL training.

• Lines 156-159 describe the randomization process. There are 3 points here:

1. 1. Please include a separate subsection for randomization per the CONSORT statement.

2. 2. The randomization and blinding have not been described in enough detail. I gathered that the design is a cluster randomized trial where the groups of students were randomized rather than individual students.

3. 3. Moreover, please clarify how you ensured equal numbers in the groups. Simple randomization does not create equal numbers in the two groups. Did you use blocking?

4. I suggest including all the details about randomization under a separate subsection per the CONSORT checklist. This would improve your manuscript readability and clarity.

• In the intervention subsection, there is confusion over what the authors exactly did in contrast to what they should or could have done. I suggest that the authors describe what they did in this study instead of what a PBL session generally looks like or going back to the setting they conducted the study in. Examples: Lines 160-169.

• Line 180: Did you do a sequential randomization procedure for this?

1. Was randomization to the cases after randomization to the pure PBL vs Hybrid PBL?

2. Since the difficulty level of the cases varied and randomization cannot balance the two groups based on 5 cases, was the between-group balance of cases controlled by any means? A better approach would probably be to ensure equal assignment of cases in the study groups rather than randomization.

3. If the groups were randomized, how did you conceal the random sequence?

• I suggest organizing the intervention into clear paragraphs describing the tutor selection and training, the instructional design, etc. in clearly distinct paragraphs to improve readability.

• I suggest the following order for your presentation to improve the flow:

1. The general format of the intervention: case-based discussion led by a tutor, the difference between the two study arms.

2. Selection and training of the tutors: who were the tutors? How did you select/ train them?

3. The cases and how they were assigned to the groups.

4. I would also like to know how the cases were developed and standardized for training purposes.

• Outcome measures: Line 193: The term target achievement rate is somehow unclear to me. The authors need to describe it before using it as a heading. I suggest using a more clear and common term instead. I suggest “students’ perceived competence in clinical reasoning”.

• The subsection was hard to follow probably due to lack of coherence within and between paragraphs. I suggest organizing this section to improve readability. One possible sequence would be: the construct, the tools to measure that construct and their reliability, and the way the tools were used in this study.

• Lines 231-233: I do not think this part adds much or helps the reader of your article understand or learn something. The authors may want to omit this. An alternative to this would be a brief statement on why the authors chose KFP and not other assessment methods of clinical reasoning.

• Lines: 237-238: Using the same disease in the KFP as discussed in the PBL session rises an issue: the KFP test may test the gained knowledge on the case during the PBL session in contrast to competence in clinical reasoning.

• Sample size: the authors have used a smaller number of participants than their calculated sample size. This may have increased the type 2 error. I suggest clarifying this with possible technical reasons why the planned number of participants could not be recruited.

• Data analysis: Since the groups were not similar at baseline, I suggest using techniques to adjust for baseline differences.

• Why did the authors only hold the focus groups among the hybrid groups? It occurred to me that the focus group sessions could have a separate effect on students’ competence in clinical reasoning via reflection or simply increased interaction with the tutors. Also, this may introduce bias as a co-intervention in the hybrid group.

Results:

• Lines 317-318: I don’t understand how a random assignment can compensate the problem. I would appreciate if the authors clarify this.

• 323-325: Statistical testing for baseline difference is discouraged by most methodologists and per the CONSORT statement. A table presenting the main participant characteristics at baseline with means/SD or number/% in each group is a suitable alternative.

• Most of the results section text contains repetition of the material in the tables. Instead, the authors had better calculate effect sizes and confidence intervals for each of main study outcomes and present that in the text. This would bold out the main results and improve readability.

Discussion:

• Using item numbers in the text as in lines 435-439 makes the text hard to follow and to understand the concepts presented. Instead, I suggest using short words that refer to the item concepts rather than numbers that require the reader to go back to check your tables several times.

• One issue with discussing individual items in your study is that the comparison of too many items without adjustment for p-value may increase the type 1 error.

• I suggest that the authors summarize and re-arrange their manuscript so that it contains shorter paragraphs. This would greatly increase the reader engagement.

• Line 492-502: Did the lecture or the focus group decrease the students’ ability to reflect? This is an important finding to discuss.

Limitations:

• First: I suggest simplifying this paragraph. eg: Although the study was randomized the study arms deferred at baseline in regards to….

• Mention the effects of co-intervention from the focus groups that may have introduced bias and what it means in the context of your findings. Since the hybrid PBL group scored lower in reflection and verbalization items after the intervention, could this be an adverse impact of the focus group or does it mean that the hybrid PBL approach decreases the students’ reflective thinking even in spite of the possible positive effects of the focus group on reflection?

Conclusion:

• Please include an overview of the main study conclusion in response to your primary question.

Reviewer 2:

The analysis of the qualitative part of the research is not very clear. Authors should explain how qualitative data were collected and how they were analyzed. Also, the coding table should be in the manuscript.

Reviewer 3:

If the authors provide the sequence of randomization in order to show if more hybrid PBL groups were requited after the Covid 19 initial phases of pandemic it might help to explain the two arms of the study differences in pre intervention self assessment.

Reviewers' comments:

Reviewer's Responses to Questions

**Comments to the Author**

1. Is the manuscript technically sound, and do the data support the conclusions?

Reviewer #1: Yes

Reviewer #2: No

Reviewer #3: Yes

2. Has the statistical analysis been performed appropriately and rigorously? 

Reviewer #1: I Don't Know

Reviewer #2: Yes

Reviewer #3: Yes

3. Have the authors made all data underlying the findings in their manuscript fully available?

Reviewer #1: Yes

Reviewer #2: Yes

Reviewer #3: Yes

4. Is the manuscript presented in an intelligible fashion and written in standard English?

Reviewer #1: No

Reviewer #2: Yes

Reviewer #3: Yes

5. Review Comments to the Author

Reviewer #1: PLOS ONE REVIEW: Hybrid PBL and Pure PBL: Which one is more effective in developing clinical reasoning skills for general medicine clerkship? - A mixed-methods randomized control trial

General Comments:

The authors aimed to test whether a hybrid PBL approach that adds a lecture session to pure PBL, enhanced the medical students’ clinical reasoning competencies compared to pure PBL. This is an important and novel question.

Their approach to the problem is particularly interesting since they have adopted a combination of a randomized controlled trial design with a qualitative focus group to assess the in-depth experience of the learners. Moreover, the authors have assessed the metacognitive aspects of clinical reasoning among their outcomes. The video supplement to the PBL sessions that included the patient’s lifestyle also grabbed my attention.

However, there are some major concerns regarding this piece of research and the way it’s presented that need to be addressed before considering the manuscript for publication in PLOS One.

• The manuscript is lengthy and this has buried its interesting and important aspects. It can be summarized to at least half its current length without sacrificing important points.

• There are major concerns about the clarity of some parts of the text.

• The authors should clarify some methodological aspects detailed below but namely

o their approach to randomization has not been described,

o the high number of outcomes that may increase the likelihood of a type I error, and how the authors have adjusted for the effects of baseline difference,

o And that the actual sample size was smaller than the calculated sample size and this may pose questions regarding the reliability of the negative findings.

Introduction:

The introduction is hard to follow and contains unnecessary explanations. I suggest summarizing it by only including a few sentences on the brief description of clinical reasoning and its significance, the components of competency in clinical reasoning, the role of PBL, and its deficits in clinical reasoning development in about two paragraphs. Followed by a third paragraph on the definition of hybrid PBL and how it can solve current problems concerning directly relevant research.

• The term “zone of proximal development” on line 99 is unclear and confusing to me. I guess the authors mean that the learners’ baseline knowledge and past experiences should be taken into account when designing a PBL session.

• There are several paragraphs in the introduction that seem redundant and only bury the main message in a lengthy discussion of unnecessary information.

Methods:

The RCT methodology is an appropriate method for the authors’ research question. However, some issues need to be addressed. The main concerns are about the randomization methods and the composition of the manuscript.

• The presentation of the methods section can be improved so that it is easier to follow. I suggest using subsections for the “participants and trial design” section: The design: where you describe the study briefly; The Participants and setting: where you describe your participants and their background including the details of the medical education system at your institution with special attention to PBL training.

• Lines 156-159 describe the randomization process. There are 3 points here:

1. 1. Please include a separate subsection for randomization per the CONSORT statement.

2. 2. The randomization and blinding have not been described in enough detail. I gathered that the design is a cluster randomized trial where the groups of students were randomized rather than individual students.

3. 3. Moreover, please clarify how you ensured equal numbers in the groups. Simple randomization does not create equal numbers in the two groups. Did you use blocking?

4. I suggest including all the details about randomization under a separate subsection per the CONSORT checklist. This would improve your manuscript readability and clarity.

• In the intervention subsection, there is confusion over what the authors exactly did in contrast to what they should or could have done. I suggest that the authors describe what they did in this study instead of what a PBL session generally looks like or going back to the setting they conducted the study in. Examples: Lines 160-169.

• Line 180: Did you do a sequential randomization procedure for this?

1. Was randomization to the cases after randomization to the pure PBL vs Hybrid PBL?

2. Since the difficulty level of the cases varied and randomization cannot balance the two groups based on 5 cases, was the between-group balance of cases controlled by any means? A better approach would probably be to ensure equal assignment of cases in the study groups rather than randomization.

3. If the groups were randomized, how did you conceal the random sequence?

• I suggest organizing the intervention into clear paragraphs describing the tutor selection and training, the instructional design, etc. in clearly distinct paragraphs to improve readability.

• I suggest the following order for your presentation to improve the flow:

1. The general format of the intervention: case-based discussion led by a tutor, the difference between the two study arms.

2. Selection and training of the tutors: who were the tutors? How did you select/ train them?

3. The cases and how they were assigned to the groups.

4. I would also like to know how the cases were developed and standardized for training purposes.

• Outcome measures: Line 193: The term target achievement rate is somehow unclear to me. The authors need to describe it before using it as a heading. I suggest using a more clear and common term instead. I suggest “students’ perceived competence in clinical reasoning”.

• The subsection was hard to follow probably due to lack of coherence within and between paragraphs. I suggest organizing this section to improve readability. One possible sequence would be: the construct, the tools to measure that construct and their reliability, and the way the tools were used in this study.

• Lines 231-233: I do not think this part adds much or helps the reader of your article understand or learn something. The authors may want to omit this. An alternative to this would be a brief statement on why the authors chose KFP and not other assessment methods of clinical reasoning.

• Lines: 237-238: Using the same disease in the KFP as discussed in the PBL session rises an issue: the KFP test may test the gained knowledge on the case during the PBL session in contrast to competence in clinical reasoning.

• Sample size: the authors have used a smaller number of participants than their calculated sample size. This may have increased the type 2 error. I suggest clarifying this with possible technical reasons why the planned number of participants could not be recruited.

• Data analysis: Since the groups were not similar at baseline, I suggest using techniques to adjust for baseline differences.

• Why did the authors only hold the focus groups among the hybrid groups? It occurred to me that the focus group sessions could have a separate effect on students’ competence in clinical reasoning via reflection or simply increased interaction with the tutors. Also, this may introduce bias as a co-intervention in the hybrid group.

Results:

• Lines 317-318: I don’t understand how a random assignment can compensate the problem. I would appreciate if the authors clarify this.

• 323-325: Statistical testing for baseline difference is discouraged by most methodologists and per the CONSORT statement. A table presenting the main participant characteristics at baseline with means/SD or number/% in each group is a suitable alternative.

• Most of the results section text contains repetition of the material in the tables. Instead, the authors had better calculate effect sizes and confidence intervals for each of main study outcomes and present that in the text. This would bold out the main results and improve readability.

Discussion:

• Using item numbers in the text as in lines 435-439 makes the text hard to follow and to understand the concepts presented. Instead, I suggest using short words that refer to the item concepts rather than numbers that require the reader to go back to check your tables several times.

• One issue with discussing individual items in your study is that the comparison of too many items without adjustment for p-value may increase the type 1 error.

• I suggest that the authors summarize and re-arrange their manuscript so that it contains shorter paragraphs. This would greatly increase the reader engagement.

• Line 492-502: Did the lecture or the focus group decrease the students’ ability to reflect? This is an important finding to discuss.

Limitations:

• First: I suggest simplifying this paragraph. eg: Although the study was randomized the study arms deferred at baseline in regards to….

• Mention the effects of co-intervention from the focus groups that may have introduced bias and what it means in the context of your findings. Since the hybrid PBL group scored lower in reflection and verbalization items after the intervention, could this be an adverse impact of the focus group or does it mean that the hybrid PBL approach decreases the students’ reflective thinking even in spite of the possible positive effects of the focus group on reflection?

Conclusion:

• Please include an overview of the main study conclusion in response to your primary question.

Reviewer #2: The analysis of the qualitative part of the research is not very clear. Authors should explain how qualitative data were collected and how they were analyzed. Also, the coding table should be in the manuscript.

Reviewer #3: If the authors provide the sequence of randomization in order to show if more hybrid PBL groups were requited after the Covid 19 initial phases of pandemic it might help to explain the two arms of the study differences in pre intervention self assessment.

6. PLOS authors have the option to publish the peer review history of their article (what does this mean?). If published, this will include your full peer review and any attached files.

Reviewer #1: No

Reviewer #2: No

Reviewer #3: **Yes: **Mohammad Hosseinzadeh

---

## [Author Response · Author response to Decision Letter 0]

5 Sep 2022

September 5, 2022

Dr. Somayeh Delavari

Title: Hybrid PBL and Pure PBL: Which one is more effective in developing clinical reasoning skills for general medicine clerkship? - A mixed-methods randomized control trial

Reference number: PONE-D-22-20122

Dear Dr. Somayeh Delavari,

Thank you for your e-mail of August 10, 2022, regarding our manuscript, “Hybrid PBL and Pure PBL: Which one is more effective in developing clinical reasoning skills for general medicine clerkship? - A mixed-methods randomized control trial”, and for the valuable comments of the reviewers. I have attached our revised manuscript, as well as a point-by-point response to the reviewers’ comments. 

We hope that the revised manuscript contains suitable responses to the comments, and we think that it has been significantly improved over the previous submission. We trust that our manuscript is now suitable for publication in PLOS ONE.

Thank you in advance for your kind consideration of our work.

Sincerely yours,

Kosuke Ishizuka, MD, PhD

Department of General Medicine, Chiba University Hospital

1-8-1, Inohana, Chuo-ku, Chiba-city, Chiba, Japan

Tel. +81-43-222-7171 (Ext. 6438); +81-43-224-4758 (Direct line)

Fax. +81-43-224-4758

E-Mail: e103007c@yokohama-cu.ac.jp

 

RESPONSES TO REVIEWER 1:

We wish to express our appreciation to the reviewer for insightful comments that have helped us to improve our paper. We hope that the revised manuscript contains suitable responses to the comments, and we think that it has been significantly improved over the previous submission. We trust that our manuscript is now suitable for publication in PLOS ONE.

NOTE

We highlighted changes of significant issues in yellow (please see “Revised Manuscript with Track Changes”).

Comments: 

General Comments:

The authors aimed to test whether a hybrid PBL approach that adds a lecture session to pure PBL, enhanced the medical students’ clinical reasoning competencies compared to pure PBL. This is an important and novel question.

Their approach to the problem is particularly interesting since they have adopted a combination of a randomized controlled trial design with a qualitative focus group to assess the in-depth experience of the learners. Moreover, the authors have assessed the metacognitive aspects of clinical reasoning among their outcomes. The video supplement to the PBL sessions that included the patient’s lifestyle also grabbed my attention.

However, there are some major concerns regarding this piece of research and the way it’s presented that need to be addressed before considering the manuscript for publication in PLOS One.

• The manuscript is lengthy and this has buried its interesting and important aspects. It can be summarized to at least half its current length without sacrificing important points.

• There are major concerns about the clarity of some parts of the text.

• The authors should clarify some methodological aspects detailed below but namely

o their approach to randomization has not been described,

o the high number of outcomes that may increase the likelihood of a type I error, and how the authors have adjusted for the effects of baseline difference,

o And that the actual sample size was smaller than the calculated sample size and this may pose questions regarding the reliability of the negative findings.

Response: 

Thank you for your general comments. 

Per your comments and amendments, we have revised and summarized our manuscript without sacrificing any important points.

We reduced the text in the previous manuscript to about half of the original text. In addition, we added some text to clarify the description of some methodological aspects such as randomization methods, techniques to adjust for baseline differences, and sample size.

We answered each of the questions you posed in the revised manuscript.

Introduction:

1) The introduction is hard to follow and contains unnecessary explanations. I suggest summarizing it by only including a few sentences on the brief description of clinical reasoning and its significance, the components of competency in clinical reasoning, the role of PBL, and its deficits in clinical reasoning development in about two paragraphs. Followed by a third paragraph on the definition of hybrid PBL and how it can solve current problems concerning directly relevant research.

Response: 

1) Along with your suggestion, we summarized the introduction section without sacrificing important points because there were several paragraphs in the introduction that seem redundant.

Changes: 

- Problem-based learning (PBL) has been introduced into graduate medical education at many medical schools as a learner-centered educational strategy to develop clinical reasoning skills [5-8]. (Introduction; lines 71 to 73.)

- However, PBL faculty have often witnessed situations where medical students got stuck on a problem and the discussion was stalled [9]. In these situations, some studies have also argued that a session with a faculty member can effectively solve the shortcomings of PBL [10, 11]. (Introduction; lines 78 to 81.)

- The purpose of this study was to compare the effectiveness of Hybrid PBL and Pure PBL in teaching clinical reasoning skills to medical students. (Introduction; lines 87 to 88.)

2) The term “zone of proximal development” on line 99 is unclear and confusing to me. I guess the authors mean that the learners’ baseline knowledge and past experiences should be taken into account when designing a PBL session.

Response: 

2) As you have indicated, the term “zone of proximal development” means that the learners’ baseline knowledge and past experiences should be taken into account when designing a PBL session.　Because the term is unclear, we revised the description of the term “zone of proximal development”. 

Changes: 

- However, PBL faculty have often witnessed situations where medical students got stuck on a problem and the discussion was stalled [9]. In these situations, some studies have also argued that a session with a faculty member can effectively solve the shortcomings of PBL [10, 11]. (Introduction; lines 78 to 81.)

- In the intervention group (Hybrid PBL group), the tutor was allowed to help the target students understand clinical discussions beyond their own medical knowledge, to help medical students problem solve, and to answer "learner-centered" questions when the target students were stuck or overlooked the key learning issues in each scenario [20, 21]. (Methods; Lines 148 to 153.)

3) There are several paragraphs in the introduction that seem redundant and only bury the main message in a lengthy discussion of unnecessary information.

Response: 

3) We summarized the introduction section without sacrificing important points because there were several paragraphs in the introduction that seem redundant.

Changes: 

- Problem-based learning (PBL) has been introduced into graduate medical education at many medical schools as a learner-centered educational strategy to develop clinical reasoning skills [5-8]. (Introduction; lines 71 to 73.)

- However, PBL faculty have often witnessed situations where medical students got stuck on a problem and the discussion was stalled [9]. In these situations, some studies have also argued that a session with a faculty member can effectively solve the shortcomings of PBL [10, 11]. (Introduction; lines 78 to 81.)

- The purpose of this study was to compare the effectiveness of Hybrid PBL and Pure PBL in teaching clinical reasoning skills to medical students. (Introduction; lines 87 to 88.)

Methods:

4) The RCT methodology is an appropriate method for the authors’ research question. However, some issues need to be addressed. The main concerns are about the randomization methods and the composition of the manuscript.

Response: 

4) Thank you for pointing it out. Besides clarifying the randomization methods, we also reorganized the method section in the following order to clarify the study flow: “Ethics statement”, “Trial design”, “Participants”, “Intervention”, “Outcome Measures (1. Student's perceived competence in PBL and satisfaction with sessions, 2. Self-evaluation of competency in clinical reasoning)”, “Sample size”, “Randomization”, “Statistical Method”, “Focus group interview”, and “Mixed methods research”. 

Changes: 

- Ethics statement. (Methods; lines 91 to 100.)

- Trial design. (Methods; lines 102 to 127.)

- Participants. (Methods; lines 129 to 144.)

- Intervention. (Methods; lines 146 to 185.)

- Outcome Measures 1) Students’ perceived competence in PBL and satisfaction with sessions. (Methods; lines 187 to 204.)

- Outcome Measures 2) Self-evaluation of competency in clinical reasoning. (Methods; lines 206 to 221.)

- Sample size. (Methods; lines 223 to 235.)

- Randomization. (Methods; lines 237 to 242.)

- Statistical Method. (Methods; lines 244 to 257.)

- Focus group interview. (Methods; lines 261 to 310.)

- Mixed methods research. (Methods; lines 312 to 315.)

5) The presentation of the methods section can be improved so that it is easier to follow. I suggest using subsections for the “participants and trial design” section: The design: where you describe the study briefly; The Participants and setting: where you describe your participants and their background including the details of the medical education system at your institution with special attention to PBL training.

Response: 

5) We changed the subsections for the “participants and trial design” subsection to “Trial design” subsection and “Participants” section.

In the “Trial design” subsection, we described this study.

In the “Participants” subsection, we described participants and background in this study, including the details of the medical education system at our institution with special attention to PBL training.

Changes: 

- Trial design

Hybrid PBL was conducted in the intervention group, and Pure PBL was conducted in the control group. The two groups were compared to examine the effectiveness of Hybrid PBL and Pure PBL in teaching clinical reasoning skills to medical students. The quantitative data in this study examined students’ perceived competence in PBL, satisfaction with sessions after educational intervention, and self-evaluation of competency in clinical reasoning before and after the educational intervention in the intervention and control groups. In the clinical clerkship, participants were being rotated in groups assigned by the university, before this study began. The groups of participants were allocated to the intervention or control groups in an equal number of groups. For the quantitative evaluation of this study, a randomized controlled trial was conducted according to the CONSORT 2010 statement (S1 Fig) [12]. To integrate quantitative and qualitative assessments, we designed a mixed-method sequential explanatory study [13-15]. This type of research study design capitalizes on the respective strengths of both quantitative and qualitative design while minimizing each other’s shortcomings. Furthermore, it allows the researchers to understand the experimental results better while incorporating the medical students’ perspectives. We intended to connect and combine the qualitative evaluation with the quantitative evaluation for our overall analysis. This is based on the advice of The National Institute of Health, which advocates a mixed-method approach to research “to improve the quality and scientific power of data” and to better address the complexity of issues in health science education today [16, 17]. Because all of the target students had experienced Pure PBL by their fourth year but not Hybrid PBL, the medical students in the intervention group were selected as the sampling group for our qualitative survey. We evaluated the advantages of learning clinical reasoning via Hybrid PBL qualitatively by focus group interview (FGI) after PBL completion to examine the cognitive aspects of the medical students, which is thought to influence the learning effectiveness of clinical reasoning by PBL. (Methods; lines 102 to 127.)

Participants

This study was conducted at a single facility in the Department of General Medicine, Chiba University Hospital in Japan. The study included 109 fifth-year medical students at the Chiba University School of Medicine who participated in a clinical clerkship at our department from January 2020 to October 2021. In the clinical clerkship, 5-6 medical students per group would rotate through the department every two weeks. A total of 20 groups rotated during the study period. Each group was exposed to PBL in one case. All subjects had already received lectures and simulation training in basic and clinical medicine by the fourth year and had passed computer-based testing and objective structured clinical examinations before their clinical clerkship. Additionally, PBL is integrated into Chiba University’s curriculum from the first to the fourth year of medical school, so all students have experienced Pure PBL. However, the subjects have no previous experience with Hybrid PBL. Students who were not able to participate in one of the conferences for any reason (sick leave or school events) were excluded from the study. We had to suspend this study due to the coronavirus disease 2019 pandemic and decided to analyze the gathered data up till then. (Methods; lines 129 to 144.)

6) Lines 156-159 describe the randomization process. There are 3 points here:

1. Please include a separate subsection for randomization per the CONSORT statement.

2. The randomization and blinding have not been described in enough detail. I gathered that the design is a cluster randomized trial where the groups of students were randomized rather than individual students.

3. Moreover, please clarify how you ensured equal numbers in the groups. Simple randomization does not create equal numbers in the two groups. Did you use blocking?

4. I suggest including all the details about randomization under a separate subsection per the CONSORT checklist. This would improve your manuscript readability and clarity.

Response: 

6) 1. We included a separate subsection for randomization per the CONSORT statement.

2. As you have indicated, the design is a cluster randomized trial where the groups of students were randomized rather than individual students.

3. In addition, we used blocking and ensured equal numbers in the groups.

4. As you mentioned, we added the separate subsection for randomization per the CONSORT statement.

Changes: 

- Randomization

The 20 eligible groups were allocated to the intervention or control groups in an equal number of groups. The groups were allocated through block randomization using Microsoft Excel (Microsoft Co., Redmond, WA, USA). The allocation was not blinded to the medical students or the faculty. The same person participated as a faculty member. (Methods; lines 237 to 242.)

7) In the intervention subsection, there is confusion over what the authors exactly did in contrast to what they should or could have done. I suggest that the authors describe what they did in this study instead of what a PBL session generally looks like or going back to the setting they conducted the study in. Examples: Lines 160-169.

Response: 

7) In the intervention section, we revised the description of what we did in this study instead of what a PBL session generally looks like or going back to the setting we conducted the study in as follows;

Changes: 

- In the control group (Pure PBL group), students learn independently under the supervision of a tutor without any intervention from a faculty member [18, 19]. In the intervention group (Hybrid PBL group), the tutor was allowed to help the target students understand clinical discussions beyond their own medical knowledge, to help medical students problem solve, and to answer "learner-centered" questions when the target students were stuck or overlooked the key learning issues in each scenario [20, 21]. (Methods; lines 147 to 153.)

8) Line 180: Did you do a sequential randomization procedure for this?

1. Was randomization to the cases after randomization to the pure PBL vs Hybrid PBL?

2. Since the difficulty level of the cases varied and randomization cannot balance the two groups based on 5 cases, was the between-group balance of cases controlled by any means? A better approach would probably be to ensure equal assignment of cases in the study groups rather than randomization.

3. If the groups were randomized, how did you conceal the random sequence?

Response: 

8) 1. Regarding these cases, we did an equal assignment of cases in the study groups rather than randomization. 

2. As you have indicated, the difficulty level of the cases varied and randomization cannot balance the two groups based on 5 cases. Therefore, we did an equal assignment of cases in the study groups rather than randomization. 

3. Regarding these cases, we assigned the cases to the study groups equally rather than via randomization.

Changes: 

- Each group was equally assigned to one of five cases in the study groups. (Methods; lines 163 to 164.)

9) I suggest organizing the intervention into clear paragraphs describing the tutor selection and training, the instructional design, etc. in clearly distinct paragraphs to improve readability.

I suggest the following order for your presentation to improve the flow:

1. The general format of the intervention: case-based discussion led by a tutor, the difference between the two study arms.

2. Selection and training of the tutors: who were the tutors? How did you select/ train them?

3. The cases and how they were assigned to the groups.

4. I would also like to know how the cases were developed and standardized for training purposes.

Response: 

9) 1-3. Along with your suggestion, we reorganized the intervention into clearly distinct paragraphs per the following order: “the general format of the intervention”, “selection and training of the tutors”, and “the cases and how we were assigned to the groups” to improve readability.

4. We added the description of how the cases were selected for training purposes.

Changes: 

- Intervention. (Methods; lines 146 to 185.)

- Each group was equally assigned to one of five cases in the study groups. The five cases were Fitz-Hugh-Curtis syndrome, panic disorder, subacute combined degeneration of the spinal cord, primary amyloidosis, and sleep apnea syndrome. All of the cases presented challenges in pattern recognition and were selected to examine analytical and diagnostic reasoning skills. In Japan, the fourth version of the national core curriculum for undergraduate medical education in 2016 newly introduced lists of possible diagnoses regarding 37 common signs, symptoms, and pathophysiology that ought to be learned as part of the six-year undergraduate curriculum. Regarding these common signs, it is required for students to acquire the competence to anticipate a set of differential diagnoses from the earliest phase of the diagnostic process, gather confirming and refuting information according to an initial hypothesis, select and perform the relevant history taking and physical examination, and interpret the findings to confirm or deny the initial hypothesis. The five cases were selected from those with any one of the 37 common signs and the diseases on five cases appearing on the Japanese National Medical Examination questions. (Methods; lines 163 to 178.)

10) Outcome measures: Line 193: The term target achievement rate is somehow unclear to me. The authors need to describe it before using it as a heading. I suggest using a more clear and common term instead. I suggest “students’ perceived competence in clinical reasoning”.

Response: 

10) Regarding outcome measures, the target achievement rates (items 1 - 5) and satisfaction with the sessions (item 6) after PBL and self-evaluation of competency in clinical reasoning (items 7 - 14) before and after the educational intervention were compared between the intervention and control groups. As you have indicated, we changed the term “target achievement rate” to “students perceived competence in PBL” in the text.

Changes: 

- Table 1. (Results; lines 341 to 346.)

- S2 Table. (Supporting information; lines 624 to 627.)

11) The subsection was hard to follow probably due to lack of coherence within and between paragraphs. I suggest organizing this section to improve readability. One possible sequence would be: the construct, the tools to measure that construct and their reliability, and the way the tools were used in this study.

Response: 

11) As you mentioned, we revised the subsection of outcome measures per the following sequence: the construct, the tools to measure that construct and their reliability, and the way the tools were used in this study.

Changes: 

- Outcome Measures 1) Students’ perceived competence in PBL and satisfaction with sessions. (Methods; lines 187 to 204.)

- Outcome Measures 2) Self-evaluation of competency in clinical reasoning. (Methods; lines 206 to 221.)

12) Lines 231-233: I do not think this part adds much or helps the reader of your article understand or learn something. The authors may want to omit this. An alternative to this would be a brief statement on why the authors chose KFP and not other assessment methods of clinical reasoning.

Lines: 237-238: Using the same disease in the KFP as discussed in the PBL session rises an issue: the KFP test may test the gained knowledge on the case during the PBL session in contrast to competence in clinical reasoning.

Response: 

12) Along with your recommendation, we removed the description about KFP test, because the number of samples assigned to each test was too few. In addition, using the same disease in the KFP as discussed in the PBL session raises an issue: the KFP test may test the gained knowledge on the case during the PBL session in contrast to competence in clinical reasoning.

13) Sample size: the authors have used a smaller number of participants than their calculated sample size. This may have increased the type 2 error. I suggest clarifying this with possible technical reasons why the planned number of participants could not be recruited.

Response: 

13) We added the description of method about the reasons for the few numbers of participants in this study.

Regarding this point, we revised the description of sample size subsection.

In addition, we added the description of limitation about this point.

Changes: 

- We had to suspend this study due to the coronavirus disease 2019 pandemic and decided to analyze the gathered data up till then. (Methods; lines 143 to 144.)

- However, we had to suspend this study due to the coronavirus disease 2019 pandemic and decided to analyze the gathered data up till then. As a result, a total of 109 students in 20 groups were considered. (Methods; lines 232 to 235.)

- Owing to the coronavirus disease 2019 pandemic, this study was suspended and resumed. Although the results should be accepted cautiously as this is a single study with relatively few participants, we believe that Hybrid PBL and Pure PBL have their own advantages in teaching medical students to acquire clinical reasoning skills. (Limitation; lines 505 to 509.)

14) Data analysis: Since the groups were not similar at baseline, I suggest using techniques to adjust for baseline differences.

Response: 

14) As you have pointed it out, we used techniques to adjust for baseline differences. Regarding self-evaluation of competency in clinical reasoning, we conducted an analysis of covariance (ANCOVA) between intervention and control groups with the self-evaluation of competency in clinical reasoning before educational intervention as a covariate and the method of PBL as a fixed factor.

In ANCOVA of the self-evaluation of competency in clinical reasoning after the educational intervention, there was no significant difference between the intervention and the control groups in all items. Meanwhile, the two-way ANOVA of the self-evaluation of competency in clinical reasoning before and after the educational intervention showed that the intervention group significantly improved in items 8. recalling appropriate differential diagnosis from the patient’s chief complaint and 14. practicing the appropriate clinical reasoning process, however, according to multiple comparisons, the control group still scored significantly higher than the intervention group in items 7. recalling appropriate history, physical examination, and tests on clinical hypothesis generation, 10. verbalizing and reflecting appropriately on own mistakes, 11. selecting keywords from the whole aspect of the patient, and 12. examining the patient while visualizing his/her daily life. The difference in results between the ANCOVA and the two-way ANOVA in self-evaluation of competency in clinical reasoning was considered to be smaller number of participants and baseline differences. We added this point to the results, discussion, and limitations section.

Changes: 

- Regarding self-evaluation of competency in clinical reasoning, we conducted an analysis of covariance (ANCOVA) between intervention and control groups before educational intervention as a covariate and the method of PBL as a fixed factor. (Methods; lines 251 to 253.)

- Regarding self-evaluation of competency in clinical reasoning, we conducted an analysis of covariance (ANCOVA) adjusted for baseline between intervention and control groups with the self-evaluation of competency in clinical reasoning before educational intervention as a covariate and the method of PBL as a fixed factor. In ANCOVA of the self-evaluation of competency in clinical reasoning after the educational intervention, there was no significant difference between the intervention and the control groups in all items (Table 3). (Results; lines 370 to 376.)

- Table 3. (Results; lines 378 to 382.)

- In ANCOVA of the self-evaluation of competency in clinical reasoning after the educational intervention, there was no significant difference between the intervention and the control groups in all items. Meanwhile, the two-way ANOVA of the self-evaluation of competency in clinical reasoning before and after the educational intervention showed that the intervention group significantly improved in items 8. recalling appropriate differential diagnosis from the patient’s chief complaint and 14. practicing the appropriate clinical reasoning process, however, according to multiple comparisons, the control group still scored significantly higher than the intervention group in items 7. recalling appropriate history, physical examination, and tests on clinical hypothesis generation, 10. verbalizing and reflecting appropriately on own mistakes, 11. selecting keywords from the whole aspect of the patient, and 12. examining the patient while visualizing his/her daily life. The difference in results between the ANCOVA and the two-way ANOVA in self-evaluation of competency in clinical reasoning was considered to be smaller number of participants and baseline differences. (Discussion; lines 440 to 454.)

- First, the differences in results between the ANCOVA and the two-way ANOVA in self-evaluation of competency in clinical reasoning were considered to be smaller number of participants and baseline differences. In the self-evaluation of competency in clinical reasoning before PBL, while the control group showed significantly higher scores for items 7. recalling appropriate history, physical examination, and tests on clinical hypothesis generation, 8. recalling appropriate differential diagnosis from the patient’s chief complaint, 9. verbalizing points that fit/don’t fit the recalled differential diagnosis appropriately, and 14. practicing the appropriate clinical reasoning process, it is possible that sampling bias may have occurred between both groups. Owing to the coronavirus disease 2019 pandemic, this study was suspended and resumed. Although the results should be accepted cautiously as this is a single study with relatively few participants, we believe that Hybrid PBL and Pure PBL have their own advantages in teaching medical students to acquire clinical reasoning skills. (Limitations; lines 496 to 509.)

15) Why did the authors only hold the focus groups among the hybrid groups? It occurred to me that the focus group sessions could have a separate effect on students’ competence in clinical reasoning via reflection or simply increased interaction with the tutors. Also, this may introduce bias as a co-intervention in the hybrid group.

Response: 

15) Because all of the target students had experienced Pure PBL by their fourth year but not Hybrid PBL, the medical students in the intervention group were selected as the sampling group for our qualitative survey. We evaluated the advantages of learning clinical reasoning via Hybrid PBL qualitatively by focus group interview (FGI) after PBL completion to examine the cognitive aspects of the medical students, which is thought to influence the learning effectiveness of clinical reasoning by PBL. 

Since the researcher was also a teacher in the class, there might be a conflict of interest between the researcher and the medical students. As a countermeasure, it was made clear to the medical students that this study would not affect their grade evaluations before the study. However, as you mentioned, the focus group sessions could have a separate effect on students’ competence in clinical reasoning via reflection or simply increased interaction with the tutors. Also, this may introduce bias as a co-intervention in the hybrid group. We added this point to the limitation section.

Changes: 

- Because all of the target students had experienced Pure PBL by their fourth year but not Hybrid PBL, the medical students in the intervention group were selected as the sampling group of qualitative survey. We evaluated the advantages of learning clinical reasoning via Hybrid PBL qualitatively by focus group interview (FGI) after PBL completion to examine the cognitive aspects of the medical students, which is thought to influence the learning effectiveness of clinical reasoning by PBL. (Methods; lines 264 to 270.)

- Fifth, we only held the focus groups among the intervention groups. Because all of the target students had experienced Pure PBL by their fourth year but not Hybrid PBL, the medical students in the intervention group were selected as the sampling group for our qualitative survey. We evaluated the advantages of learning clinical reasoning via Hybrid PBL qualitatively by focus group interview (FGI) after PBL completion to examine the cognitive aspects of the medical students, which is thought to influence the learning effectiveness of clinical reasoning by PBL. In addition, since the researcher was also a teacher in the class, there might be a conflict of interest. As a countermeasure, it was made clear to the medical students that this study would not affect their grade evaluations before the study. However, the focus group sessions could have a separate effect on students’ competence in clinical reasoning via reflection or simply increased interaction with the tutors. Also, this may introduce bias as a co-intervention in the intervention group. (Limitations; lines 524 to 537.)

Results:

16) Lines 317-318: I don’t understand how a random assignment can compensate the problem. I would appreciate if the authors clarify this.

Response: 

16) We described in our previous manuscript that “Owing to the coronavirus disease 2019 pandemic, this study was suspended and resumed. Yet, because of the random assignment, the data were still considered acceptable for analysis.” However, as you have pointed it out, a random assignment cannot compensate the problem. Therefore, we removed the description about this point.

In addition, we added the description of limitation section.

Changes: 

- Owing to the coronavirus disease 2019 pandemic, this study was suspended and resumed. Although the results should be accepted cautiously as this is a single study with relatively few participants, we believe that Hybrid PBL and Pure PBL have their own advantages in teaching medical students to acquire clinical reasoning skills. (Limitations; lines 505 to 509.)

17) 323-325: Statistical testing for baseline difference is discouraged by most methodologists and per the CONSORT statement. A table presenting the main participant characteristics at baseline with means/SD or number/% in each group is a suitable alternative.

Response: 

17) As you mentioned, we added the table presenting the main participant characteristics at baseline with mean/SD in Table 2.

Changes: 

- Table 2 showed the baseline of self-evaluation of competency in clinical reasoning before educational intervention between the intervention and the control groups. (Results; lines 349 to 350.)

- Table 2. (Results; lines 364 to 368.)

18) Most of the results section text contains repetition of the material in the tables. Instead, the authors had better calculate effect sizes and confidence intervals for each of main study outcomes and present that in the text. This would bold out the main results and improve readability.

Response: 

18) We calculated effect sizes and confidence intervals for each of the main study outcomes and presented them in the table.

Changes: 

- Regarding self-evaluation of competency in clinical reasoning, we conducted an analysis of covariance (ANCOVA) adjusted for baseline between intervention and control groups with the self-evaluation of competency in clinical reasoning before educational intervention as a covariate and the method of PBL as a fixed factor. In ANCOVA of the self-evaluation of competency in clinical reasoning after the educational intervention, there was no significant difference between the intervention and the control groups in all items (Table 3). (Results; lines 370 to 376.)

- Table 3. (Results; lines 378 to 382.)

Discussion:

19) Using item numbers in the text as in lines 435-439 makes the text hard to follow and to understand the concepts presented. Instead, I suggest using short words that refer to the item concepts rather than numbers that require the reader to go back to check your tables several times.

Response: 

19) To improve the readability of the text, we revised the discussion to expand on the item numbers in the text. 

Changes: 

- To examine whether the self-evaluation of competency in clinical reasoning changed between the intervention and control groups, we conducted a two-way ANOVA (Table 4), where the within-subjects factor was the time of implementation of PBL, and the between-subjects factor was the method of PBL. As a result, a significant interaction effect was shown in item 8. recalling appropriate differential diagnosis from the patient’s chief complaint (F(1,97)=5.295, p=0.024) and item 14. practicing the appropriate clinical reasoning process (F(1,97)=4.016, p=0.038). There was no significant interaction effect between the intervention and the control groups in items 7. recalling appropriate history, physical examination, and tests on clinical hypothesis generation, 9. verbalizing points that fit/don’t fit the recalled differential diagnosis appropriately, 10. verbalizing and reflecting appropriately on own mistakes, 11. selecting keywords from the whole aspect of the patient, 12. examining the patient while visualizing his/her daily life, and 13. considering biological, psychological, and social perspectives. According to multiple comparisons, the control group showed significantly better results in item 7. recalling appropriate history, physical examination, and tests on clinical hypothesis generation (F(1,97)=6.796, p=0.011), item 10. verbalizing and reflecting appropriately on own mistakes (F(1,97)=4.352, p=0.040), item 11. selecting keywords from the whole aspect of the patient (F(1,97)=5.607, p=0.020), and item 12. examining the patient while visualizing his/her daily life (F(1,97)=7.120, p=0.009). In addition, the intervention and control groups showed significant improvement in self-evaluation of competency in clinical reasoning after the implementation of PBL. (Results; lines 384 to 404.)

- Meanwhile, the two-way ANOVA of the self-evaluation of competency in clinical reasoning before and after the educational intervention showed that the intervention group significantly improved in items 8. recalling appropriate differential diagnosis from the patient’s chief complaint and 14. practicing the appropriate clinical reasoning process. However, according to multiple comparisons, the control group still scored significantly higher than the intervention group in items 7. recalling appropriate history, physical examination, and tests on clinical hypothesis generation, 10. verbalizing and reflecting appropriately on own mistakes, 11. selecting keywords from the whole aspect of the patient, and 12. examining the patient while visualizing his/her daily life. The difference in results between the ANCOVA and the two-way ANOVA in self-evaluation of competency in clinical reasoning was considered to be smaller number of participants and baseline differences. (Discussion; lines 443 to 454.)

- The two-way ANOVA of the self-evaluation of competency in clinical reasoning before and after the educational intervention showed that the intervention group significantly improved in items 8. recalling appropriate differential diagnosis from the patient’s chief complaint and 14. practicing the appropriate clinical reasoning process. (Discussion; lines 459 to 463.)

- Based on the two-way ANOVA of self-evaluation of competency in clinical reasoning, the control group scored higher than the intervention group in items 7. recalling appropriate history, physical examination, and tests on clinical hypothesis generation, 10. verbalizing and reflecting appropriately on own mistakes, 11. selecting keywords from the whole aspect of the patient, and 12. examining the patient while visualizing his/her daily life. (Discussion; lines 479 to 484.)

- In the self-evaluation of competency in clinical reasoning before PBL, while the control group showed significantly higher scores for items 7. recalling appropriate history, physical examination, and tests on clinical hypothesis generation, 8. recalling appropriate differential diagnosis from the patient’s chief complaint, 9. verbalizing points that fit/don’t fit the recalled differential diagnosis appropriately, and 14. practicing the appropriate clinical reasoning process, it is possible that sampling bias may have occurred between both groups. (Limitations; lines 498 to 505).

20) One issue with discussing individual items in your study is that the comparison of too many items without adjustment for p-value may increase the type 1 error.

Response: 

20) Regarding self-evaluation of competency in clinical reasoning, we added an analysis of covariance (ANCOVA) adjusted for baseline.

In ANCOVA of the self-evaluation of competency in clinical reasoning after the educational intervention, there was no significant difference between the intervention and the control groups in all items. Meanwhile, the two-way ANOVA of the self-evaluation of competency in clinical reasoning before and after the educational intervention showed that the intervention group significantly improved in items 8. recalling appropriate differential diagnosis from the patient’s chief complaint and 14. practicing the appropriate clinical reasoning process, however, according to multiple comparisons, the control group still scored significantly higher than the intervention group in items 7. recalling appropriate history, physical examination, and tests on clinical hypothesis generation, 10. verbalizing and reflecting appropriately on own mistakes, 11. selecting keywords from the whole aspect of the patient, and 12. examining the patient while visualizing his/her daily life. The difference in results between the ANCOVA and the two-way ANOVA in self-evaluation of competency in clinical reasoning was considered to be smaller number of participants and baseline differences. We added the description about this point to results, discussion, and limitations.

Changes: 

- Regarding self-evaluation of competency in clinical reasoning, we conducted an analysis of covariance (ANCOVA) between intervention and control groups before educational intervention as a covariate and the method of PBL as a fixed factor. (Methods; lines 251 to 253.)

- Regarding self-evaluation of competency in clinical reasoning, we conducted an analysis of covariance (ANCOVA) adjusted for baseline between intervention and control groups with the self-evaluation of competency in clinical reasoning before educational intervention as a covariate and the method of PBL as a fixed factor. In ANCOVA of the self-evaluation of competency in clinical reasoning after the educational intervention, there was no significant difference between the intervention and the control groups in all items (Table 3). (Results; lines 370 to 376.)

- Table 3. (Results; lines 378 to 382.)

- In ANCOVA of the self-evaluation of competency in clinical reasoning after the educational intervention, there was no significant difference between the intervention and the control groups in all items. Meanwhile, the two-way ANOVA of the self-evaluation of competency in clinical reasoning before and after the educational intervention showed that the intervention group significantly improved in items 8. recalling appropriate differential diagnosis from the patient’s chief complaint and 14. practicing the appropriate clinical reasoning process, however, according to multiple comparisons, the control group still scored significantly higher than the intervention group in items 7. recalling appropriate history, physical examination, and tests on clinical hypothesis generation, 10. verbalizing and reflecting appropriately on own mistakes, 11. selecting keywords from the whole aspect of the patient, and 12. examining the patient while visualizing his/her daily life. The difference in results between the ANCOVA and the two-way ANOVA in self-evaluation of competency in clinical reasoning was considered to be smaller number of participants and baseline differences. (Discussion; lines 440 to 454.)

- First, the differences in results between the ANCOVA and the two-way ANOVA in self-evaluation of competency in clinical reasoning were considered to be smaller number of participants and baseline differences. In the self-evaluation of competency in clinical reasoning before PBL, while the control group showed significantly higher scores for items 7. recalling appropriate history, physical examination, and tests on clinical hypothesis generation, 8. recalling appropriate differential diagnosis from the patient’s chief complaint, 9. verbalizing points that fit/don’t fit the recalled differential diagnosis appropriately, and 14. practicing the appropriate clinical reasoning process, it is possible that sampling bias may have occurred between both groups. Owing to the coronavirus disease 2019 pandemic, this study was suspended and resumed. Although the results should be accepted cautiously as this is a single study with relatively few participants, we believe that Hybrid PBL and Pure PBL have their own advantages in teaching medical students to acquire clinical reasoning skills. (Limitations; lines 496 to 509.)

21) I suggest that the authors summarize and re-arrange their manuscript so that it contains shorter paragraphs. This would greatly increase the reader engagement.

Response: 

21) Along with your suggestion, we summarized and rearranged our manuscript so that it contains shorter paragraphs.

Changes: 

- Discussion. (Discussion; lines 436 to 492.)

22) Line 492-502: Did the lecture or the focus group decrease the students’ ability to reflect? This is an important finding to discuss.

Response: 

22) Yes, we considered that the lecture might decrease the students’ ability to reflect because the FGI was conducted after filling out the post-PBL Questionnaires.

We added the timing of conducting FGI exactly.

Changes: 

- One faculty member administered the FGI to a group of 5-6 medical students immediately after filling out the post-PBL questionnaires. (Methods; lines 279 to 280.)

We thank the reviewer for such pertinent comments. We hope that the revised manuscript contains suitable responses to the comments, and we think that it has been significantly improved compared to the previous submission. We trust that our manuscript is now suitable for publication in PLOS ONE.

 

RESPONSES TO REVIEWER 2:

We wish to express our appreciation to the reviewer for insightful comments that have helped us to improve our paper. We hope that the revised manuscript contains suitable responses to the comments, and we think that it has been significantly improved over the previous submission. We trust that our manuscript is now suitable for publication in PLOS ONE.

NOTE

We highlighted changes of significant issues in yellow (please see “Revised Manuscript with Track Changes”).

Comments: 

The analysis of the qualitative part of the research is not very clear. Authors should explain how qualitative data were collected and how they were analyzed. Also, the coding table should be in the manuscript.

Response: 

Thank you for your comments.

Along with the consolidated criteria for reporting qualitative research (COREQ), we revised the qualitative analysis part of the research and clarified how qualitative data were collected and how they were analyzed.

In addition, we added the coding table in the manuscript.

Changes: 

- Qualitative evaluation was conducted following the quantitative evaluation to assess the acquisition of higher-order intellectual skills, and the results were integrated as a mixed-method sequential explanatory study [13-15]. Because all of the target students had experienced Pure PBL by their fourth year but not Hybrid PBL, the medical students in the intervention group were selected as the sampling group for our qualitative survey. We evaluated the advantages of learning clinical reasoning via Hybrid PBL qualitatively using focus group interview (FGI) after PBL completion to examine the cognitive aspects of the medical students, which is thought to influence the learning effectiveness of clinical reasoning by PBL. After obtaining informed consent from medical students in the intervention group, we conducted an FGI face-to-face immediately after completion of the PBL, which lasted about 10 minutes to minimize participants’ fatigue. In the FGI, the advantages of Hybrid PBL in improving clinical reasoning skills were investigated. The aims of the study were reviewed at the beginning of each focus group. The interview content was discussed among the interviewers (HT, KS), and an interview guide was developed (S3 Table). Medical students were asked the following open-ended questions: "What are the advantages of Hybrid PBL? Why do you think so?” The responses of the focus groups were recorded and transcribed verbatim. One faculty member administered the FGI to a group of 5-6 medical students immediately after filling out the post-PBL questionnaires. The FGI sampling was 52 students (10 groups) from all intervention groups. The FGI interviewers were trained facilitators from the faculty (KI, KS, YH, YY, JK, SU, YY, DY, ES, NH, SY, TT, KN) who were in charge of PBL. They had experience in higher education in their respective countries and had previously conducted educational research. The FGI interviewers did not share their personal attitudes and behaviors. A team debrief was held after each focus group session, and field notes were documented. There were no repeat interviews. Participants did not provide feedback nor review transcripts. (Methods; lines 262 to 288.)

- For the qualitative research, content analysis was used to analyze the themes. A preliminary analytic template, aligned with the focus group guide, was developed as a starting point for analysis. Two researchers (KI, HT) independently read and did the initial coding of all focus group transcripts. Thereafter, to ensure the quality of the research, researcher triangulation was conducted, in which two researchers (KI, HT) discussed, identified, and agreed on the coding of the descriptors. Following the coding, similar codes were grouped into categories and sub-categories as they emerged from the data. The categories and subcategories were regularly discussed and reviewed by KS (who had experience in qualitative research) to ensure credibility of the findings. Cohen's kappa coefficient was used to evaluate the intra-rater validity between the two researchers [24]. The intra-rater validity was evaluated using the kappa coefficient (0.8-1.0=almost perfect, 0.6-0.8=substantial, 0.4-0.6=moderate, 0.2-0.4=good). The consolidated criteria for reporting qualitative research (COREQ) checklist was used to report our findings [25]. (Methods; lines 289 to 302.)

- To examine the effects of Hybrid PBL on the cognitive processes of medical students, the analytic categories were set according to the six cognitive process dimensions of the revised Bloom's taxonomy [26]. After open coding, similar codes were classified into subcategories and categories. We analyzed the concepts in each dimension of the six cognitive process dimensions of the revised Bloom's taxonomy and calculated the number of analysis units for each concept [26]. We also grouped similar codes as themes and checked which dimension of the cognitive process dimension they correspond to. (Methods; lines 303 to 310.)

- To integrate quantitative and qualitative assessments, a mixed methods study was conducted with an exploratory sequential design [13-15]. The qualitative assessment was intended to support and explain the results of the quantitative assessment. (Methods; lines 313 to 315.)

- Table 5. (Results; lines 433 to 434.)

We thank the reviewer for such pertinent comments. We hope that the revised manuscript contains suitable responses to the comments and has been significantly improved compared to the previous submission. We trust that our manuscript is now suitable for publication in PLOS ONE.

 

RESPONSES TO REVIEWER 3:

We wish to express our appreciation to the reviewer for insightful comments that have helped us to improve our paper. We hope that the revised manuscript contains suitable responses to the comments, and we think that it has been significantly improved over the previous submission. We trust that our manuscript is now suitable for publication in PLOS ONE.

NOTE

We highlighted changes of significant issues in yellow (please see “Revised Manuscript with Track Changes”).

Comments: 

If the authors provide the sequence of randomization in order to show if more hybrid PBL groups were requited after the Covid 19 initial phases of pandemic it might help to explain the two arms of the study differences in pre intervention self assessment.

Response: 

Thank you for your comments.

Although we confirmed this point, more hybrid PBL groups were not requited after the initial phases of COVID-19 pandemic.

In addition, we used techniques to adjust for baseline differences. Regarding self-evaluation of competency in clinical reasoning, we conducted an analysis of covariance (ANCOVA) between intervention and control groups with the self-evaluation of competency in clinical reasoning before educational intervention as a covariate and the method of PBL as a fixed factor.

As you mentioned, we summarized the methods and results section and clarified the description such as the sequence of randomization.

In ANCOVA of the self-evaluation of competency in clinical reasoning after the educational intervention, there was no significant difference between the intervention and the control groups in all items. Meanwhile, the two-way ANOVA of the self-evaluation of competency in clinical reasoning before and after the educational intervention showed that the intervention group significantly improved in items 8. recalling appropriate differential diagnosis from the patient’s chief complaint and 14. practicing the appropriate clinical reasoning process, however, according to multiple comparisons, the control group still scored significantly higher than the intervention group in items 7. recalling appropriate history, physical examination, and tests on clinical hypothesis generation, 10. verbalizing and reflecting appropriately on own mistakes, 11. selecting keywords from the whole aspect of the patient, and 12. examining the patient while visualizing his/her daily life. The difference in results between the ANCOVA and the two-way ANOVA in self-evaluation of competency in clinical reasoning was considered to be smaller number of participants and baseline differences. We added the description about this point to the results, discussion, and limitations section.

Changes: 

- Methods. (Methods; lines 90 to 315.)

- Regarding self-evaluation of competency in clinical reasoning, we conducted an analysis of covariance (ANCOVA) between intervention and control groups before educational intervention as a covariate and the method of PBL as a fixed factor. (Methods; lines 251 to 253.)

- Regarding self-evaluation of competency in clinical reasoning, we conducted an analysis of covariance (ANCOVA) adjusted for baseline between intervention and control groups with the self-evaluation of competency in clinical reasoning before educational intervention as a covariate and the method of PBL as a fixed factor. In ANCOVA of the self-evaluation of competency in clinical reasoning after the educational intervention, there was no significant difference between the intervention and the control groups in all items (Table 3). (Results; lines 370 to 376.)

- Table 3. (Results; lines 378 to 382.)

- In ANCOVA of the self-evaluation of competency in clinical reasoning after the educational intervention, there was no significant difference between the intervention and the control groups in all items. Meanwhile, the two-way ANOVA of the self-evaluation of competency in clinical reasoning before and after the educational intervention showed that the intervention group significantly improved in items 8. recalling appropriate differential diagnosis from the patient’s chief complaint and 14. practicing the appropriate clinical reasoning process, however, according to multiple comparisons, the control group still scored significantly higher than the intervention group in items 7. recalling appropriate history, physical examination, and tests on clinical hypothesis generation, 10. verbalizing and reflecting appropriately on own mistakes, 11. selecting keywords from the whole aspect of the patient, and 12. examining the patient while visualizing his/her daily life. The difference in results between the ANCOVA and the two-way ANOVA in self-evaluation of competency in clinical reasoning was considered to be smaller number of participants and baseline differences. (Discussion; lines 440 to 454.)

- First, the differences in results between the ANCOVA and the two-way ANOVA in self-evaluation of competency in clinical reasoning were considered to be smaller number of participants and baseline differences. In the self-evaluation of competency in clinical reasoning before PBL, while the control group showed significantly higher scores for items 7. recalling appropriate history, physical examination, and tests on clinical hypothesis generation, 8. recalling appropriate differential diagnosis from the patient’s chief complaint, 9. verbalizing points that fit/don’t fit the recalled differential diagnosis appropriately, and 14. practicing the appropriate clinical reasoning process, it is possible that sampling bias may have occurred between both groups. Owing to the coronavirus disease 2019 pandemic, this study was suspended and resumed. Although the results should be accepted cautiously as this is a single study with relatively few participants, we believe that Hybrid PBL and Pure PBL have their own advantages in teaching medical students to acquire clinical reasoning skills. (Limitations; lines 496 to 509.)

We thank the reviewer for such pertinent comments. We hope that the revised manuscript contains suitable responses to the comments, and we think that it has been significantly improved compared to the previous submission. We trust that our manuscript is now suitable for publication in PLOS ONE.

---

## [Decision Letter · Decision Letter 1]

28 Sep 2022

PONE-D-22-20122R1Hybrid PBL and Pure PBL: Which one is more effective in developing clinical reasoning skills for general medicine clerkship? - A mixed-methods randomized control trialPLOS ONE

Dear Dr. Ishizuka,

Thank you for submitting your manuscript to PLOS ONE. After careful consideration, we feel that it has merit but does not fully meet PLOS ONE’s publication criteria as it currently stands. Therefore, we invite you to submit a revised version of the manuscript that addresses the points raised during the review process.

We look forward to receiving your revised manuscript.

Kind regards,

Somayeh Delavari, Ph.D.,

Academic Editor

PLOS ONE

Additional Editor Comments (if provided):

Reviewer 5

In this manuscript, the block randomization method was used. According to the type of study, which is an RCT, this method needs to be explained more, such as how many blocks are determined and how permutation was done.

Reviewer 1

I appreciate the authors for their considerable efforts in revising their paper. It has tremendously improved. However, there are a few remaining concerns that should be addressed before the manuscript is suitable for publication in PONE.

1. The most significant point is that the manuscript, although reduced in size, is still lengthy and this can affect readability. I suggest summarizing some data presentations in tables instead of presenting every detail in the texts, not repeating methods in the results section, and getting help from a professional editor to cut redundancies. The limitations section is specifically too long for its messages and can be summarized.

2. Regarding the randomization procedure, I still find it confusing. I assume that the authors have done cluster randomization. If that is the case, this should be edited in the paper title. Moreover, the authors explanation brings about confusions regarding allocation concealment in contrast to blinding which are two distinct matters in an RCT design. I suggest getting help from their RCT methodologist on how they can best present their methods.

3. Regarding randomization, was there a random sequence list for inclusion of students? If so, how was it kept from the researchers figuring out the sequence(allocation concealment)?

4. I suggest that a statistician confirms that the un-adjusted p-values used by the authors are appropriate given multiple comparisons.

5. The authors say they have designed their study based on the CONSORT statement. The CONSORT statement is intended to improve the reporting quality and is not about designing your study.

Reviewers' comments:

Reviewer's Responses to Questions

**Comments to the Author**

1. If the authors have adequately addressed your comments raised in a previous round of review and you feel that this manuscript is now acceptable for publication, you may indicate that here to bypass the “Comments to the Author” section, enter your conflict of interest statement in the “Confidential to Editor” section, and submit your "Accept" recommendation.

Reviewer #1: (No Response)

Reviewer #2: All comments have been addressed

Reviewer #4: (No Response)

Reviewer #5: All comments have been addressed

2. Is the manuscript technically sound, and do the data support the conclusions?

Reviewer #1: Yes

Reviewer #2: Yes

Reviewer #4: Yes

Reviewer #5: Yes

3. Has the statistical analysis been performed appropriately and rigorously? 

Reviewer #1: I Don't Know

Reviewer #2: Yes

Reviewer #4: Yes

Reviewer #5: Yes

4. Have the authors made all data underlying the findings in their manuscript fully available?

Reviewer #1: Yes

Reviewer #2: Yes

Reviewer #4: Yes

Reviewer #5: Yes

5. Is the manuscript presented in an intelligible fashion and written in standard English?

Reviewer #1: Yes

Reviewer #2: Yes

Reviewer #4: Yes

Reviewer #5: Yes

6. Review Comments to the Author

Reviewer #1: I appreciate the authors for their considerable efforts in revising their paper. It has tremendously improved. However, there are a few remaining concerns that should be addressed before the manuscript is suitable for publication in PONE.

1. The most significant point is that the manuscript, although reduced in size, is still lengthy and this can affect readability. I suggest summarizing some data presentations in tables instead of presenting every detail in the texts, not repeating methods in the results section, and getting help from a professional editor to cut redundancies. The limitations section is specifically too long for its messages and can be summarized.

2. Regarding the randomization procedure, I still find it confusing. I assume that the authors have done cluster randomization. If that is the case, this should be edited in the paper title. Moreover, the authors explanation brings about confusions regarding allocation concealment in contrast to blinding which are two distinct matters in an RCT design. I suggest getting help from their RCT methodologist on how they can best present their methods.

3. Regarding randomization, was there a random sequence list for inclusion of students? If so, how was it kept from the researchers figuring out the sequence(allocation concealment)?

4. I suggest that a statistician confirms that the un-adjusted p-values used by the authors are appropriate given multiple comparisons.

5. The authors say they have designed their study based on the CONSORT statement. The CONSORT statement is intended to improve the reporting quality and is not about designing your study.

Reviewer #2: The steps of qualitative research are not well explained. Please specify the steps of qualitative research with numbers.

Reviewer #4: (No Response)

Reviewer #5: In this manuscript, the block randomization method was used. According to the type of study, which is an RCT, this method needs to be explained more, such as how many blocks are determined and how permutation was done.

7. PLOS authors have the option to publish the peer review history of their article (what does this mean?). If published, this will include your full peer review and any attached files.

Reviewer #1: No

Reviewer #2: No

Reviewer #4: No

Reviewer #5: No

---

## [Author Response · Author response to Decision Letter 1]

29 Oct 2022

October 29, 2022

Dr. Somayeh Delavari

Title: Hybrid PBL and Pure PBL: Which one is more effective in developing clinical reasoning skills for general medicine clerkship? - A mixed-methods randomized control trial

【Revised title: Hybrid PBL and Pure PBL: Which one is more effective in developing clinical reasoning skills for general medicine clerkship? - A mixed-methods cluster randomized controlled trial】

Reference number: PONE-D-22-20122.R1

Dear Dr. Somayeh Delavari,

Thank you for your e-mail of September 29, 2022, regarding our manuscript, “Hybrid PBL and Pure PBL: Which one is more effective in developing clinical reasoning skills for general medicine clerkship? - A mixed-methods randomized control trial”, and for the valuable comments of the reviewers. In order to reflect the design of the study accurately, we changed the title to “Hybrid PBL and Pure PBL: Which one is more effective in developing clinical reasoning skills for general medicine clerkship? - A mixed-methods cluster randomized controlled trial”. I have attached our revised manuscript, as well as a point-by-point response to the reviewers’ comments. 

We hope that the revised manuscript contains suitable responses to the comments, and we think that it has been significantly improved over the previous submission. We trust that our manuscript is now suitable for publication in PLOS ONE.

Thank you in advance for your kind consideration of our work.

Sincerely yours,

Kosuke Ishizuka, MD, PhD

Department of General Medicine, Chiba University Hospital

1-8-1, Inohana, Chuo-ku, Chiba-city, Chiba, Japan

Tel. +81-43-222-7171 (Ext. 6438); +81-43-224-4758 (Direct line)

Fax. +81-43-224-4758

E-Mail: e103007c@yokohama-cu.ac.jp

 

RESPONSES TO REVIEWER 1:

We wish to express our appreciation to the reviewer for insightful comments that have helped us to improve our paper. We hope that the revised manuscript contains suitable responses to the comments, and we think that it has been significantly improved over the previous submission. We trust that our manuscript is now suitable for publication in PLOS ONE.

NOTE

We highlighted changes of significant issues in yellow (please see “Revised Manuscript with Track Changes”).

Comments: 

I appreciate the authors for their considerable efforts in revising their paper. It has tremendously improved. However, there are a few remaining concerns that should be addressed before the manuscript is suitable for publication in PONE.

Response: 

Thank you for your comments. 

Per your comments and amendments, we have revised and summarized our manuscript without sacrificing any important points.

We have also answered each of the questions you posed in the revised manuscript.

1) The most significant point is that the manuscript, although reduced in size, is still lengthy and this can affect readability. I suggest summarizing some data presentations in tables instead of presenting every detail in the texts, not repeating methods in the results section, and getting help from a professional editor to cut redundancies. The limitations section is specifically too long for its messages and can be summarized.

Response: 

Per your suggestion, we have further summarized the indicated sections without compromising the intended meaning. We have also obtained help from a professional editor to further cut any redundancies. We reduced the number of words in the text from 4913 words in the previous manuscript to 4243 words in the revised manuscript. Please find a certificate of proofreading in English attached.

Other reviewer has pointed out that we should specify the steps of qualitative research with numbers. Therefore, we added this point as Table 1 so as not to avoid increasing the number of words. 

Changes: 

- Text. (Text; lines 64 to 497.)

- Table 1. (Methods; line 283.)

- Participant characteristics. (Results; lines 305 to 310.)

- Table 2. (Results; line 312.)

- 1) Students’ perceived competence in PBL and satisfaction with sessions. (Results; lines 314 to 324.)

- 2) Self-evaluation of competency in clinical reasoning. (Results; lines 326 to 383.)

- Content analysis. (Results; lines 385 to 401.)

- Limitations. (Limitations; lines 459 to 488.)

2) Regarding the randomization procedure, I still find it confusing. I assume that the authors have done cluster randomization. If that is the case, this should be edited in the paper title. Moreover, the authors explanation brings about confusions regarding allocation concealment in contrast to blinding which are two distinct matters in an RCT design. I suggest getting help from their RCT methodologist on how they can best present their methods.

3) Regarding randomization, was there a random sequence list for inclusion of students? If so, how was it kept from the researchers figuring out the sequence (allocation concealment)?

Response: 

Thank you for pointing it out.

To clarify any possible confusion, we conducted a cluster randomized controlled trial in this study, not block randomized controlled trial. We have thus changed the paper title to “Hybrid PBL and Pure PBL: Which one is more effective in developing clinical reasoning skills for general medicine clerkship? - A mixed-methods cluster randomized controlled trial” to better reflect this. 

We got help from methodologist and revised our manuscript in order to clarify our methods. Regarding randomization, there was not a random sequence list for inclusion of students. The study included 109 fifth-year medical students at the Chiba University School of Medicine who participated in a clinical clerkship at our department from January 2020 to October 2021. This study was embedded into a clinical clerkship rotation in the department; thus, the participants were not sampled randomly. In the clinical clerkship, 5-6 medical students per group were rotated through the department every two weeks. A total of 20 groups were rotated during the study period. They were allocated to the intervention or control groups equally through cluster randomization using Microsoft Excel (Microsoft Co., Redmond, WA, USA). The allocation was not blinded to the medical students or the faculty. We have added this to our manuscript.

In addition, we have also added to the limitation section the possibility of bias using this method. Specifically, subjective bias may have affected the results. In addition, the assignment of these groups by the university before the start of this study might also have been biased, which may have affected the study results.

Changes: 

- Title.

- This study was embedded into a clinical clerkship rotation in the department; thus, the participants were not sampled randomly. (Methods; lines 118 to 120.)

- Randomization

The 20 eligible groups were allocated to the intervention or control groups in an equal number of groups. The groups were allocated through cluster randomization using Microsoft Excel (Microsoft Co., Redmond, WA, USA). The allocation was not blinded to the medical students or the faculty. (Methods; lines 219 to 223.)

- Sixth, the allocation to the two groups could not be blinded to the participants and the faculty members, and subjective bias may have affected the results. Furthermore, the assignment of these groups by the university before the start of this study might also have been biased, which may have affected the study results. (Limitations; lines 485 to 488.)

4) I suggest that a statistician confirms that the un-adjusted p-values used by the authors are appropriate given multiple comparisons.

Response: 

Along with your recommendation, we asked the statistician to perform the statistical analysis for check and confirmed that the un-adjusted p-values we used are appropriate given multiple comparisons.

We performed a two-way repeated measure analysis of variance (ANOVA), where the within-subjects factor was the time of implementation and the between-subjects factor was the method of PBL because there are two factors. In this statistical analysis, it should be described as a two-way repeated measure ANOVA exactly, which we have revised accordingly. 

Additional statistical analysis of the normal distribution showed that it did not follow a normal distribution. Therefore, we added to the limitation section the possibility of alpha error in two-way ANOVA.

Changes: 

- In two-way repeated measure analysis of variance, self-evaluation of competency in clinical reasoning was significantly improved in the intervention group in "recalling appropriate differential diagnosis from patient's chief complaint" (F(1,97)=5.295, p=0.024) and "practicing the appropriate clinical reasoning process" (F(1,97)=4.016, p=0.038). (Abstract; lines 46 to 50.)

- In addition, to examine whether the self-evaluation of competency in clinical reasoning changed between the intervention and control groups, we conducted a two-way repeated measure analysis of variance (ANOVA), where the within-subjects factor was the time of implementation, and the between-subjects factor was the method of PBL. (Methods; lines 234 to 238.)

- In addition, an alpha error may have occurred in the statistical analysis of the two-way ANOVA because it did not follow a normal distribution. (Limitations; lines 464 to 466.)

5) The authors say they have designed their study based on the CONSORT statement. The CONSORT statement is intended to improve the reporting quality and is not about designing your study.

Response: 

As you have indicated, we have revised the description of CONSORT statement as follows:

Changes: 

- To improve the reporting quality, we conducted a cluster randomized controlled trial based on the CONSORT 2010 statement (S1 Fig) [12]. (Methods; lines 106 to 108.)

We thank the reviewer for such pertinent comments. We hope that the revised manuscript contains suitable responses to the comments, and we think that it has been significantly improved compared to the previous submission. We trust that our manuscript is now suitable for publication in PLOS ONE.

 

RESPONSES TO REVIEWER 2:

We wish to express our appreciation to the reviewer for insightful comments that have helped us to improve our paper. We hope that the revised manuscript contains suitable responses to the comments, and we think that it has been significantly improved over the previous submission. We trust that our manuscript is now suitable for publication in PLOS ONE.

NOTE

We highlighted changes of significant issues in yellow (please see “Revised Manuscript with Track Changes”).

Comments: 

The steps of qualitative research are not well explained. Please specify the steps of qualitative research with numbers.

Response: 

Thank you for pointing it out.

Along with your comments, we clarified the steps of qualitative research with numbers.

Other reviewer has pointed out that we should reduce the number of text word count. Therefore, we added this point as Table 1 so as not to avoid increasing the number of words.

Changes: 

- For the qualitative research, content analysis was used to analyze the themes (Table 1) [22, 23]. (Methods; lines 270 to 271.)

- Table 1. (Methods; line 283.)

We thank the reviewer for such pertinent comments. We hope that the revised manuscript contains suitable responses to the comments, and we think that it has been significantly improved compared to the previous submission. We trust that our manuscript is now suitable for publication in PLOS ONE.

 

RESPONSES TO REVIEWER 5:

We wish to express our appreciation to the reviewer for insightful comments that have helped us to improve our paper. We hope that the revised manuscript contains suitable responses to the comments, and we think that it has been significantly improved over the previous submission. We trust that our manuscript is now suitable for publication in PLOS ONE.

NOTE

We highlighted changes of significant issues in yellow (please see “Revised Manuscript with Track Changes”).

Comments: 

In this manuscript, the block randomization method was used. According to the type of study, which is an RCT, this method needs to be explained more, such as how many blocks are determined and how permutation was done.

Response: 

Thank you for your comments.

To clarify any possible confusion, we conducted a cluster randomized controlled trial in this study, not a block randomized controlled trial. We have thus changed the paper title to “Hybrid PBL and Pure PBL: Which one is more effective in developing clinical reasoning skills for general medicine clerkship? - A mixed-methods cluster randomized controlled trial” to better reflect this. 

We got help from methodologist and revised our manuscript in order to clarify our methods. The study included 109 fifth-year medical students at the Chiba University School of Medicine who participated in a clinical clerkship at our department from January 2020 to October 2021. This study was embedded into a clinical clerkship rotation in the department; thus, the participants were not sampled randomly. In the clinical clerkship, 5-6 medical students per group were rotated through the department every two weeks. A total of 20 groups were rotated during the study period. They were allocated to the intervention or control groups equally through cluster randomization using Microsoft Excel (Microsoft Co., Redmond, WA, USA). The allocation was not blinded to the medical students or the faculty. We have added this to our manuscript.

In addition, along with the other reviewer comments, we have also added to the limitation section the possibility of bias using this method.

Changes: 

- Title.

- This study was embedded into a clinical clerkship rotation in the department; thus, the participants were not sampled randomly. (Methods; lines 118 to 120.)

- Randomization

The 20 eligible groups were allocated to the intervention or control groups in an equal number of groups. The groups were allocated through cluster randomization using Microsoft Excel (Microsoft Co., Redmond, WA, USA). The allocation was not blinded to the medical students or the faculty. (Methods; lines 219 to 223.)

- Sixth, the allocation to the two groups could not be blinded to the participants and the faculty members, and subjective bias may have affected the results. Furthermore, the assignment of these groups by the university before the start of this study might also have been biased, which may have affected the study results. (Limitations; lines 485 to 488.)

We thank the reviewer for such pertinent comments. We hope that the revised manuscript contains suitable responses to the comments, and we think that it has been significantly improved compared to the previous submission. We trust that our manuscript is now suitable for publication in PLOS ONE.

---

## [Decision Letter · Decision Letter 2]

12 Dec 2022

Hybrid PBL and Pure PBL: Which one is more effective in developing clinical reasoning skills for general medicine clerkship? - A mixed-methods cluster randomized controlled trial

PONE-D-22-20122R2

Dear Dr. Ishizuka,

We’re pleased to inform you that your manuscript has been judged scientifically suitable for publication and will be formally accepted for publication once it meets all outstanding technical requirements.

Kind regards,

Somayeh Delavari, Ph.D.,

Academic Editor

PLOS ONE

---

## [Editor Report · Acceptance letter]

12 Jan 2023

PONE-D-22-20122R2 

Hybrid PBL and Pure PBL: Which one is more effective in developing clinical reasoning skills for general medicine clerkship? - A mixed-method study 

Dear Dr. Ishizuka:

I'm pleased to inform you that your manuscript has been deemed suitable for publication in PLOS ONE. Congratulations! Your manuscript is now with our production department. 

Kind regards, 

on behalf of

Dr. Somayeh Delavari 

Academic Editor

PLOS ONE